# Flexible Modeling of Diversity with Strongly Log-Concave Distributions

**Joshua Robinson**
Massachusetts Institute of Technology
joshrob@mit.edu

**Suvrit Sra**
Massachusetts Institute of Technology
suvrit@mit.edu

**Stefanie Jegelka**
Massachusetts Institute of Technology
stefje@csail.mit.edu

## Abstract

Strongly log-concave (SLC) distributions are a rich class of discrete probability distributions over subsets of some ground set. They are strictly more general than strongly Rayleigh (SR) distributions such as the well-known determinantal point process. While SR distributions offer elegant models of diversity, they lack an easy control over how they express diversity. We propose SLC as the right extension of SR that enables easier, more intuitive control over diversity, illustrating this via examples of practical importance. We develop two fundamental tools needed to apply SLC distributions to learning and inference: *sampling* and *mode finding*. For sampling we develop an MCMC sampler and give theoretical mixing time bounds. For mode finding, we establish a weak log-submodularity property for SLC functions and derive optimization guarantees for a distorted greedy algorithm.

## 1  Introduction

A variety of machine learning tasks involve selecting diverse subsets of items. How we model diversity is, therefore, a key concern with possibly far-reaching consequences. Recently popular probabilisitic models of diversity include determinantal point processes [32, 39], and more generally, strongly Rayleigh (SR) distributions [8, 35]. These models have been successfully deployed for subset selection in applications such as video summarization [44], fairness [13], model compression [46], anomaly detection [49], the Nyström method [41], generative models [24, 40], and accelerated coordinate descent [51]. While valuable and broadly applicable, SR distributions have one main drawback: it is difficult to control the strength and nature of diversity they model.

We counter this drawback by leveraging strongly log-concave (SLC) distributions [3–5]. These distributions are strictly more general than SR measures, and possess key properties that enable easier, more intuitive control over diversity. They derive their name from SLC polynomials introduced by Gurvits already a decade ago [30]. More recently they have shot into prominence due to their key role in developing deep connections between discrete and continuous convexity, with subsequent applications in combinatorics [1, 10, 33]. In particular, they lie at the heart of recent breakthrough results such as a proof of Mason's conjecture [4] and obtaining a fully polynomial-time approximation scheme for counting the number of bases of arbitrary matroids [3, 5]. We remark that all these works assume *homogeneous* SLC polynomials.

We build on this progress to develop fundamental tools for general SLC distributions, namely, *sampling* and *mode finding*. We highlight the flexibility of SLC distributions through two settings of importance in practice: (i) raising any SLC distribution to a power $\alpha \in [0, 1]$; and (ii) incorporating a constraint that allows sampling sets of *any* size up to a budget. In contrast to similar modifications

to SR measures (see e.g., [49]), these settings retain the crucial SLC property. Setting (i) allows us to conveniently tune the strength of diversity by varying a single parameter; while setting (ii) offers greater flexibility than *fixed* cardinality distributions such as a $k$-determinantal point process [38]. This observation is simple yet important, especially since the "right" value of $k$ is hard to fix *a priori*.

**Contributions.** We briefly summarize the main contributions of this work below.

- We introduce the class of strongly log-concave distributions to the machine learning community, showing how it can offer a flexible discrete probabilistic model for distributions over subsets.
- We prove various closure properties of SLC distributions (Theorems 2-5), and show how to use these properties for better controlling the distributions used for inference.
- We derive sampling algorithms for SLC and related distributions, and analyze their corresponding mixing times both theoretically and empirically (Algorithm 1, Theorem 8).
- We study the negative dependence of SLC distributions by deriving a weak log-submodularity property (Theorem 11). Optimization guarantees for a selection of greedy algorithms are obtained as a consequence (Theorem 12).

As noted above, our results build on the remarkable recent progress in [3–5] and [10]. The biggest difference between the previous work and this work is our focus on general non-homogeneous SLC polynomials, corresponding to distributions over sets of varying cardinality, as opposed to purely the homogeneous, i.e., fixed-cardinality, case. This broader focus necessitates development of some new machinery, because unlike SR polynomials, the class of SLC polynomials is not closed under homogenization. We summarize the related work below for additional context.

## 1.1 Related work

**SR polynomials.** Strongly Rayleigh distributions were introduced in [8] as a class of discrete distributions possessing several strong negative dependence properties. It did not take long for their potential in machine learning to be identified [39]. Particular attention has been paid to determinantal point processes due to the intuitive way they capture negative dependence, and the fact that they are parameterized by a single positive semi-definite kernel matrix. Convenient parameterization has allowed an abundance of fast algorithms for learning the kernel matrix [23, 26, 45, 47], and sampling [2, 42, 50]. SR distributions are a fascinating and elegant probabilistic family whose applicability in machine learning is still an emerging topic [17, 35, 43, 48].

**SLC polynomials.** Gurvits introduced SLC polynomials a decade ago [30] and studied their connection to discrete convex geometry. Recently this connection was significantly developed [10, 5] by establishing that matroids, and more generally M-convex sets, are characterized by the strong log-concavity of their generating polynomial. This is in contrast to SR, for which it is known that some matroids have generating polynomials that are *not* SR [9].

**Log-submodular distributions.** Distributions over subsets that are log-submodular (or supermodular) are amenable to mode finding and variational inference with approximation guarantees, by exploiting the optimization properties of submodular functions [20–22]. Theoretical bounds on sampling time require additional assumptions [29]. Iyer and Bilmes [34] analyze inference for submodular distributions, establishing polynomial approximation bounds.

**MCMC samplers and mixing time.** The seminal works [18, 19] offer two tools for obtaining mixing time bounds for Markov chains: lower bounding the spectral gap, or log-Sobolev constant. These techniques have been successfully deployed to obtain mixing time bounds for homogenous SR distributions [2], general SR distributions [42], and recently homogenous SLC distributions [5].

## 2  Background and setup

**Notation.** We write $[n] = \{1, \ldots, n\}$, and denote by $2^{[n]}$ the power set $\{S \mid S \subseteq [n]\}$. For any variable $u$, write $\partial_u$ to denote $\frac{\partial}{\partial u}$; in case $u = z_i$, we often abbreviate further by writing $\partial_i$ instead of $\partial_{z_i}$. For $S \subseteq [n]$ and $\alpha \in \mathbb{N}^n$ let $\mathbf{1}_S \in \{0,1\}^n$ denote the binary indicator vector of $S$, and define $|\alpha| = \sum_{i=1}^{n} \alpha_i$. We also write variously $\partial_z^S = \prod_{i \in S} \partial_i$ and $\partial_z^\alpha = \prod_{i \in [n]} \partial_i^{\alpha_i}$ where $\alpha_i = 0$ means we do not take any derivatives with respect to $z_i$. We let $z^S$ and $z^\alpha$ denote the monomials $\prod_{i \in S} z_i$ and $\prod_{i=1}^{n} z_i^{\alpha_i}$ respectively. For $K = \mathbb{R}$ or $\mathbb{R}_+$ we write $K[z_1, \ldots, z_n]$ to denote the set of all polynomials in the variables $z = (z_1, \ldots, z_n)$ whose coefficients belong to $K$. A polynomial is

said to be $d$-homogeneous if it is the sum of monomials *all* of which are of degree $d$. Finally, for a set $X$ we shall minimize clutter by using $X \cup i$ and $X \setminus i$ to denote $X \cup \{i\}$ and $X \setminus \{i\}$ respectively.

**SLC distributions.** We consider distributions $\pi : 2^{[n]} \to [0,1]$ on the subsets of a ground set $[n]$. There is a one-to-one correspondence between such distributions, and their generating polynomials

$$f_\pi(z) := \sum_{S \subseteq [n]} \pi(S) \prod_{i \in S} z_i = \sum_{S \subseteq [n]} \pi(S) z^S. \tag{1}$$

The central object of interest in this paper is the class of strongly log-concave distributions, which is defined by imposing certain log-concavity requirements on the corresponding generating polynomials.

**Definition 1.** A polynomial $f \in \mathbb{R}_+[z_1, \ldots, z_n]$ is *strongly log-concave* (SLC) if every derivative of $f$ is log-concave. That is, for any $\alpha \in \mathbb{N}^n$ either $\partial^\alpha f = 0$, or the function $\log(\partial^\alpha f(z))$ is concave at all $z \in \mathbb{R}_+^n$. We say a distribution $\pi$ is strongly log-concave if its generating polynomial $f_\pi$ is strongly log-concave; we also say $\pi$ is $d$-homogeneous if $f_\pi$ is $d$-homogeneous.

There are many examples of SLC distributions; we note a few important ones below.

- Determinantal point processes [39, 27, 38, 41], and more generally, Strongly Rayleigh (SR) distributions [8, 17, 43, 35].
- Exponentiated (for exponents in $[0,1)$) homogeneous SR distributions [49, 5].
- The uniform distribution on the independent sets of a matroid [4].

SR distributions satisfy several strong negative dependence properties (e.g., log-submodularity and negative association). The fact that SLC is a strict superset of SR suggests that SLC distributions possess some weaker negative dependence properties. These properties will play a crucial role in the two fundamental tasks that we study in this paper: *sampling* and *mode finding*.

**Sampling.** Our first task is to efficiently draw samples from an SLC distribution $\pi$. To that end, we seek to develop Markov Chain Monte Carlo (MCMC) samplers whose mixing time (see Section 4 for definition) can be well-controlled. For homogeneous $\pi$, the breakthrough work of Anari et al. [5] provides the first analysis of fast-mixing for a simple Markov chain called `Base Exchange Walk`; this analysis is further refined in [15]. `Base Exchange Walk` is defined as follows: if currently at state $S \subseteq [n]$, remove an element $i \in S$ uniformly at random. Then move to $R \supset S \setminus \{i\}$ with probability proportional to $\pi(R)$. This describes a transition kernel $Q(S, R)$ for moving from $S$ to $R$. We build on these works to obtain the first mixing time bounds for sampling from *general* (i.e., not necessarily homogeneous) SLC distributions (Section 4).

**Mode finding.** Our second main goal is optimization, where we consider the more general task of finding a mode of an SLC distribution subject to a cardinality constraint. This task involves solving $\max_{|S| \leq d} \pi(S)$. This task is known to be NP-hard even for SR distributions; indeed, the maximum volume subdeterminant problem [14] is a special case. We consider a more practical approach based on observing that SLC distributions satisfy a relaxed notion of log-submodularity, which enables us to adapt simple greedy algorithms.

Before presenting the details about sampling and optimization, we need to first establish some key theoretical properties of general SLC distributions. This is the subject of the next section.

## 3 Theoretical tools for general SLC polynomials

In this technical section we develop the theory of strong log-concavity by detailing several transformations of an SLC polynomial $f$ that preserve strong log-concavity. Such closure properties can be essential for proving the SLC property, or for developing algorithmic results. Due to the correspondence between distributions on $2^{[n]}$ and their generating polynomials, each statement concerning polynomials can be translated into a statement about probability distributions. The forthcoming results assume polynomials that are supported on the independent sets of a matroid. This can be viewed as a minor technical assumption since, to the best of our knowledge, all known SLC polynomials are supported on the independent sets of a matroid. A fundamental correspondence between homogenous SLC distributions and bases of a matroid was observed in [10], however it remains an open question to precisely understand this relationship for non-homogenous SLC polynomials. The following theorem is a crucial stepping stone to sampling from non-homogeneous SLC distributions, and to sampling with cardinality constraints.

**Theorem 2.** *Let* $f = \sum_{S \subseteq [n]} c_S z^S \in \mathbb{R}_+[z_1, \ldots, z_n]$ *be SLC, and suppose the support of the sum is the collection of independent sets of a rank $d$ matroid. Then for any $k \le d$ the following polynomial is SLC:*

$$\mathcal{H}_k f(z, y) = \sum_{|S| \le k} \frac{c_S}{(k - |S|)!} z^S y^{k - |S|}.$$

The above operation is also referred to as *scaled homogenization*, since the resulting polynomial is homogeneous and there is an added $1/(k - |S|)!$ factor. In fact, we may extend Theorem 2 to allowing the user to add an additional exponentiating factor:

**Theorem 3.** *Let* $f = \sum_{S \subseteq [n]} c_S z^S \in \mathbb{R}_+[z_1, \ldots, z_n]$ *be SLC, and suppose the support of the sum is the collection of independent sets of a rank $d$ matroid. Then for $0 \le \alpha \le 1$ and any $k \le d$ the following polynomial is SLC:*

$$\mathcal{H}_{k,\alpha} f(z, y) = \sum_{|S| \le k} \frac{c_S^\alpha}{(k - |S|)!} z^S y^{k - |S|}.$$

Notably, Theorem 3 fails for *all* $\alpha > 1$. For a proof of this see Appendix A.2.

Next, we show that polarization preserves strong log-concavity. Polarization essentially means to replace a variable with a higher power by multiple "copies", each occurring only with power one, in a way that the resulting polynomial is symmetric (or permutation-invariant) in those copies. This is achieved by averaging over elementary symmetric polynomials. Formally, the polarization of the polynomial $f = \sum_{|S| \le d} c_S z^S y^{d - |S|} \in \mathbb{R}[z_1, \ldots, z_n, y]$ is defined to be

$$\Pi f(z_1, \ldots, z_n, y_1, \ldots, y_d) = \sum_{|S| \le d} c_S z^S \binom{d}{|S|}^{-1} e_{d - |S|}(y_1, \ldots, y_d)$$

where $e_k(y_1, \ldots, y_d)$ is the $k$th elementary symmetric polynomial in $d$ variables. The polarization $\Pi f$ has the following three properties:

1. It is symmetric in the variables $y_1, \ldots, y_d$;
2. Setting $y_1 = \ldots = y_d = y$ recovers $f$;
3. $\Pi f$ is multiaffine, and hence the generating polynomial of a distribution on $2^{[n+d]}$.

Closure under polarization, combined with the homogenization results (Theorems 2 and 3) allows non-homogeneous distributions to be transformed into homogenous ones. This allows general SLC distributions to be transformed into homogenous SLC distributions for which fast mixing results are known [5]. How to work backwards to obtain samples from the original distribution will be the topic of the next section.

**Theorem 4.** [1] *Let* $f = \sum_{S \subseteq [n]} c_S z^S y^{d - |S|} \in \mathbb{R}_+[z_1, \ldots, z_n, y]$ *be SLC, and the support of the sum is the collection of independent sets of a rank $d$ matroid. Then the polarization $\Pi f$ is SLC.*

Putting all of the preceding results together we obtain the following important corollary. It is this observation that will allow us to do mode finding for SLC distributions and exponentiated, cardinality constrained SLC distributions.

**Corollary 5.** *Let* $f = \sum_{S \subseteq [n]} c_S z^S \in \mathbb{R}_+[z_1, \ldots, z_n]$ *be SLC, and suppose the support of the sum is the collection of independent sets of a rank $d$ matroid. Then $\Pi(\mathcal{H}_{k,\alpha} f)$ is SLC for any $k \le d$ and $0 \le \alpha \le 1$.*

In Appendix A.4 we also show that SLC distributions are closed under conditioning on a fixed set size. We mention those results since they may be of independent interest, but omit them from the main text since we do not use them further in this paper.

# 4    Sampling from strongly log-concave distributions

In this section we outline how to use the SLC closure results from Section 3 to build a sampling algorithm for general SLC distributions and prove mixing time bounds. Recall that we are considering a probability distribution $\pi : 2^{[n]} \to [0,1]$ that is strongly log-concave. The mixing time of a Markov chain $(Q, \pi)$ started at $S_0$ is $t_{S_0}(\varepsilon) = \min\{t \in \mathbb{N} \mid \left\|Q^t(S_0, \cdot) - \pi\right\|_1 \le \varepsilon\}$ where $Q^t$ is the $t$-step transition kernel. For the remainder of this section we consider the distribution $\nu$ where $\nu(S) \propto \pi(S)^\alpha \mathbf{1}\{|S| \le d\}$ for $0 \le \alpha \le 1$, and $d \in [n]$. In particular, this includes $\pi$ itself. The power $\alpha$ allows to vary the degree of diversity induced by the distribution: $\alpha < 1$ smooths $\nu$, making it less diverse. Indeed, as $\alpha \to 0$, $\nu$ converges to the uniform distribution, which promotes no diversity. Meanwhile $\alpha > 1$ (although outside the scope of our results) makes $\nu$ more pointy, with $\nu$ collapsing to a point mass as $\alpha \to \infty$.

Our strategy is as follows: we first "extend" $\nu$ to a distribution $\nu_{\mathrm{sh}}$ over subsets of size $|n|$ of $[n+d]$ to obtain a homogeneous distribution. If we can sample from $\nu_{\mathrm{sh}}$, then we can extract a sample $S \subseteq [n]$ of a scaled version of $\nu$ by simply restricting a sample $T \sim \nu_{\mathrm{sh}}$ to $T \cap [n]$. If $\nu$ was SR, then $\nu_{\mathrm{sh}}$ would also be SR, and a fast sampler follows from this observation [42]. But, for general SLC distributions (and their powers), $\nu_{\mathrm{sh}}$ is not SLC, and deriving a sampler is more challenging.

To still enable the homogenization strategy, we instead derive a carefully scaled version of a homogeneous version of $\nu$ that, as we prove, is homogeneneous and SLC and hence tractable. We use this rescaled version as a proposal distribution in a sampler for $\nu_{\mathrm{sh}}$. To obtain an appropriately scaled extended, homogeneous variant $\nu$, we first translate Corollary 5 into probabilistic language.

**Theorem 6.** *Suppose that the support of the sum in the generating polynomial of $\nu$ is the collection of independent sets of a rank $d$ matroid. Then for any $k \le d$ the following probability distribution on $2^{[n+k]}$ is SLC:*

$$\mathcal{H}_k\nu(S) \propto \begin{cases} \binom{k}{|S \cap [n]|}^{-1} \frac{\nu(S \cap [n])}{(k - |S \cap [n]|)!}, & \textit{for all } S \subseteq [n+k] \textit{ such that } |S| = k \\ 0, & \textit{otherwise.} \end{cases}$$

*Proof.* Observe that the generating polynomial of $\mathcal{H}_k\nu$ is $\Pi(\mathcal{H}_k f)$ where $f$ denotes the generating polynomial of $\nu$. The result follows immediately from Corollary 5. □

The ultimate proposal that we use is not $\mathcal{H}_k\nu$, but a modified version $\mu$ that better aligns with $\nu$:

$$\mu(S) \propto \left(\frac{d}{e}\right)^{d - |S \cap [n]|} \mathcal{H}_d\nu(S).$$

**Proposition 7.** *If $\nu$ is SLC, then $\mu$ is SLC.*

*Proof.* Lemma 39 in the Appendix says that strong log-concavity is preserved under linear transformations of the coordinates. This implies that $\mu$ is SLC since its generating polynomial is $\Pi((\mathcal{H}_d f) \circ T)$ where $f$ is the generating polynomial of $\nu$ and $T$ is the linear transform defined by: $y \mapsto \frac{d}{e}y$ and $z_i \mapsto z_i$ for $i = 1 \dots, n$. □

Importantly, since $\mu$ is homogeneous and SLC, the `Base Exchange Walk` for $\mu$ mixes rapidly. Let $Q$ denote the Markov transition kernel for `Base Exchange Walk` on $2^{[n+d]}$ for $\mu$. We use $Q$ as a proposal, and then compute the appropriate acceptance probability to obtain a chain that mixes to the symmetric homogenization $\nu_{\mathrm{sh}}$ of $\nu$. The target $\nu_{\mathrm{sh}}$ is a $d$-homogenous distribution on $2^{[n+d]}$:

$$\nu_{\mathrm{sh}}(S) \propto \binom{d}{|S \cap [n]|}^{-1} \nu(S \cap [n]), \text{ for all } S \subseteq [n+d] \text{ such that } |S| = d.$$

A crucial property of $\nu_{\mathrm{sh}}$ is that its marginalization over the "dummy" variables yields $\nu$, i.e., $\sum_{T : T \cap [n] = S} \nu_{\mathrm{sh}}(T) = \nu(S)$. Therefore, after obtaining a sample $T \sim \nu_{\mathrm{sh}}$ one then obtains a sample from $\nu$ by computing $T \cap [n]$.

It is a simple computation to show that the acceptance probabilities in Algorithm 1 are indeed the Metropolis-Hastings acceptance probabilities for sampling from $\nu_{\mathrm{sh}}$ using the proposal $Q$. Therefore

---
**Algorithm 1** Metropolis-Hastings sampler for $\nu_{\text{sh}}$ with proposal $Q$
---
 1: Initialize $S \subseteq [n+d]$
 2: **while** not mixed **do**
 3:     Set $k \leftarrow \left| S \cap [n] \right|$
 4:     Propose move $T \sim Q(S, \cdot)$
 5:     **if** $\left| T \cap [n] \right| = k - 1$ **then**
 6:         $R \leftarrow T$ with probability $\min\{1, \frac{e}{d}(d-k+1)\}$, otherwise stay at $S$
 7:     **if** $\left| T \cap [n] \right| = k$ **then**
 8:         $R \leftarrow T$
 9:     **if** $\left| T \cap [n] \right| = k + 1$ **then**
10:         $R \leftarrow T$ with probability $\min\{1, \frac{d}{e}\frac{1}{(d-k)}\}$, otherwise stay at $S$
---

the chain mixes to $\nu_{\text{sh}}$. We obtain the following mixing time bound, recalling that the mixing time of $(Q, \nu_{\text{sh}})$ is $t_{S_0}(\varepsilon) = \min\{t \in \mathbb{N} \mid \left\| Q^t(S_0, \cdot) - \nu_{\text{sh}} \right\|_1 \leq \varepsilon\}$.

**Theorem 8.** *For $d \geq 8$ the mixing time of the chain in Algorithm 1 started at $S_0$ satisfies the bound*

$$t_{S_0}(\varepsilon) \leq \frac{1}{e\sqrt{2\pi}} d^{5/2} 2^d \left( \log\log\left\{ \binom{d}{|S_0|} \frac{1}{\nu(S_0)} \right\} + \log\frac{1}{2\varepsilon^2} \right).$$

A similar bound holds for $d < 8$. We note that although the mixing time bound scales poorly in $d$, the bound has the interesting property of being independent of the ground set size $n$. Furthermore, the bound is meaningful since the total number of subsets of $n$ objects of size $d$ is $\sum_{j=0}^{d} \binom{n}{j} = \Omega(2^n)$ if $d \geq n/2 - \sqrt{n}$ and equal to $\Omega(2^{n/2})$ if $d \leq n/2 - \sqrt{n}$, [36]. So the mixing time bound is exponentially better than brute force. Later we will detail experiments that suggest that this bound is loose in $d$.

**Efficient implementation.** It is sufficient to only maintain $R = S \cap [n]$ since $\nu_{\text{sh}}$ is exchangeable in the variables $\{n+1, \ldots, n+d\}$. Sampling $T \sim Q(S, \cdot)$ involves dropping $i \in S$ uniformly at random, then computing the probability of $\mu((S \setminus i) \cup j)$ for each $j$ not in $S \setminus i$. However again, by the exchangeability of $\mu$ in $\{n+1, \ldots, n+d\}$ this probability is the same for each $j$ in $\{n+1, \ldots, n+d\}$ and so only needs to be performed for one such $j$.

## 5    Maximization of weakly log-submodular functions

In this section we explore the negative dependence properties of SLC functions (unnormalized SLC distributions) through the lens of submodularity: a well known negative dependence property [8]. In an earlier version of this paper we conjectured that SLC functions have the strong property of log-submodularity. This conjecture has been disproved in a recent note [28].

**Proposition 9** (Propositions 1 and 2 [28]). *The the distribution with generating polynomial*

$$f(x, y, z) = \frac{1}{22}\left(4 + 3(x+y+z) + 3(xy + xz + yz)\right)$$

*is SLC but not log-submodular.*

In response, we introduce a new notion of weak submodularity and show that any function $\nu$ such that $\mathcal{H}_d\nu$ is SLC is weakly log-submodular. Finally, we prove that a distorted greedy optimization procedure leads to optimization guarantees for weak (log-)submodular functions for the cardinality constrained problem OPT $\in \arg\max_{|S| \leq k} \nu(S)$. Appendix C contains similar results for constrained greedy optimization of *increasing* weak (log-)submodular functions and unconstrained double greedy optimization of non-negative (log-)submodular functions.

**Definition 10.** We call a function $\rho : 2^{[n]} \to \mathbb{R}$ $\gamma$-weakly submodular if for any $S \subseteq [n]$ and $i, j \in [n] \setminus S$ with $i$ and $j$ not equal, we have

$$\rho(S) + \rho(S \cup \{i, j\}) \leq \gamma + \rho(S \cup i) + \rho(S \cup j).$$

We say $\nu : 2^{[n]} \to \mathbb{R}_+$ is $\gamma$-weakly log-submodular if $\log \nu$ is $(\log \gamma)$-weakly submodular.

When $\gamma = 0$ this reduces to the classic notion of submodularity. Note carefully that our notion of weak submodularity differs from a notion of weak submodularity that already appears in the literature [16, 31, 37]. Building on a result by Brändén and Huh [10], we prove the following result.

**Theorem 11.** *Any non-negative function $\rho : 2^{[n]} \to \mathbb{R}_+$ with support contained in $\{S \subseteq [n] : |S| \leq d\}$ and generating polynomial $f$ such that $\mathcal{H}_d f$ is strongly log-concave is $\gamma$-weakly log-submodular for $\gamma = 4\left(1 - \frac{1}{d}\right)$.*

Next we show how weak log-submodularity gives a path to optimizing strongly log-concave functions. Consider $\rho : 2^{[n]} \to \mathbb{R}$, assumed to be $\gamma$-weakly submodular. Note in particular we do *not* assume that $\rho$ is non-negative. This is important since we are interested in applying this procedure to the logarithm of a distribution, which need not be non-negative. Define $c_e = \max\{\rho([n] \setminus e) - \rho([n]), 0\}$, and $c(S) = \sum_{e \in S} c_e$. We use the convention that $c(\varnothing) = 0$. Then we may decompose $\rho = \eta - c$ where $\eta = \rho + c$. Note that $\eta$ is $\gamma$-weakly submodular and $c$ is a non-negative function.

We will extend the distorted greedy algorithm by [25, 31] to our notion of weak submodularity. To do so, we introduce the distorted objective $\Phi_i(S) = (1 - 1/k)^{k-i} \eta(S) - c(S)$ for $i = 0, \ldots k$. The distorted greedy algorithm greedily builds a set $R$ of size at most $d$ by forming a sequence $\varnothing = S_0, S_1, \ldots, S_{k-1}, S_k = R$ such that $S_{i+1}$ is formed by adding the element $e_i \in [n]$ to $S_i$ that maximizes $\Phi_{i+1}(S_i \cup e_i) - \Phi_{i+1}(S_i)$ so long as the increment is positive.

---

**Algorithm 2** Distorted greedy weak submodular constrained maximization of $\nu = \eta - c$

---

1: Let $S_0 = \varnothing$
2: **for** $i = 0, \ldots, k - 1$ **do**
3:     Set $e_i = \arg\max_{e \in [n]} \Phi_{i+1}(S_i \cup e) - \Phi_{i+1}(S_i)$
4:     **if** $\Phi_{i+1}(S_i \cup e_i) - \Phi_{i+1}(S_i) > 0$ **then**
5:         $S_{i+1} \leftarrow S_i \cup e_i$
6:     **else** $S_{i+1} \leftarrow S_i$
7: **return** $R = S_k$

---

**Theorem 12.** *Suppose $\rho : 2^{[n]} \to \mathbb{R}$ is $\gamma$-weakly submodular and $\rho(\varnothing) = 0$. Then the solution $R = S_k$ obtained by the distorted greedy algorithm satisfies*

$$\rho(R) = \eta(R) - c(R) \geq \left(1 - \frac{1}{e}\right)\left(\eta(OPT) - \frac{1}{2}\ell(\ell - 1)\gamma\right) - c(OPT),$$

*where $OPT \in \arg\max_{|S| \leq k} \rho(S)$ and $\ell := |OPT| \leq k$.*

Note any weakly submodular function can be brought into the required form by subtracting $\rho(\varnothing)$ if it is non-zero. If $\nu$ is weakly log-submodular, we can decompose $\nu = \eta/c$ such that $\log \eta$ and $\log c$ perform the same role as $\eta$ and $c$ did in the weakly submodular setting. Then by applying Theorem 12 to $\log \nu$ we obtain the following corollary.

**Corollary 13.** *Suppose $\nu : 2^{[n]} \to \mathbb{R}_+$ is $\gamma$-weakly log-submodular and $\nu(\varnothing) = 1$. Then the solution $R = S_k$ obtained by the distorted greedy algorithm satisfies*

$$\nu(R) = \frac{\eta(R)}{c(R)} \geq \gamma^{-\frac{1}{2}\ell(\ell-1)(1-1/e)} \frac{\eta(OPT)^{1-1/e}}{c(OPT)}.$$

## 6 Experiments

In this section we empirically evaluate the mixing time of Algorithm 1. We use the standard *potential scale reduction factor* metric to measure convergence to the stationary distribution [11]. The method involves running several chains in parallel and computing the average variance within each chain and between the chains. The PSRF score is the ratio of the between variance over the within variance and is usually above 1. When the PSRF score is close to 1 then the chains are considered to be mixed. In all of our experiments we run three chains in parallel and declare them to be mixed once the PSRF score drops below 1.05.

Figure 1 considers the results of running the Metropolis-Hastings algorithm on a sequence of problems with different cardinality constraints $d$. In each case we considered the distribution

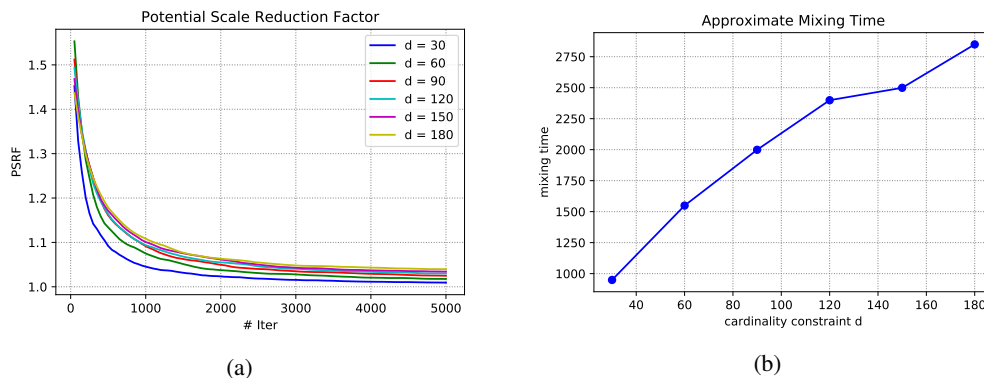

(a)                                                (b)

Figure 1: Empirical mixing time analysis for sampling a ground set of size $n = 250$ and various cardinality constraints $d$, (a) the PSRF score for each set of chains, (b) the approximate mixing time obtained by thresholding at PSRF equal to $1.05$.

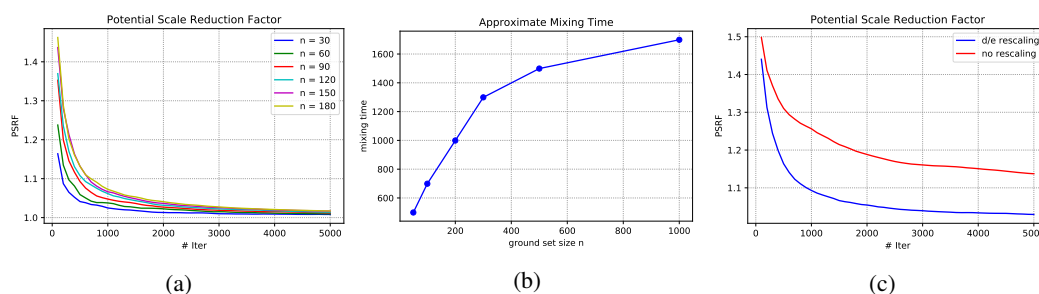

(a)                               (b)                               (c)

Figure 2: (a,b) Empirical mixing time analysis for sampling a set of size at most $d = 40$ for varying ground set sizes, (a) the PSRF score for each set of chains, (b) the approximate mixing time obtained by thresholding at PSRF equal to $1.05$, (c) comparison of Algorithm 1 and a M-H algorithm where the proposal is built using $\mathcal{H}_d\nu$: $d = 100$ and $n = 250$.

$\nu(S) \propto \sqrt{\det(L_S)}\mathbf{1}\{|S| \leq d\}$ where $L$ is a randomly generated $250 \times 250$ PSD matrix. Here $L_S$ denotes the $|S| \times |S|$ submatrix of $L$ whose indices belong to $S$. These simulations suggest that the mixing time grows linearly in $d$ for a fixed $n$.

Figure 2 considers the results of running the Metropolis-Hastings algorithm on a sequence of problems with different ground set sizes. In each case we considered the distribution $\nu(S) \propto \sqrt{\det(L_S)}\mathbf{1}\{|S| \leq 40\}$ where $L$ is a randomly generated PSD matrix where of appropriate size $n$. These simulations suggest that the mixing time grows sublinearly in $n$ for a fixed $d$.

It is important to know whether the mixing time is robust to different spectra $\sigma_L$ of $L$. We consider three cases, (i) smooth decay $\sigma_L = [n]$, (ii) a single large eigenvalue $\sigma_L = \{n, (n-1)/2, (n-2)/2, \ldots, 2/2, 1/2\}$, and (iii) one fifth of the eigenvalues are equal to $n$, the rest equal to $1/n$. Note that due to normalization, multiplying the spectrum by a constant does not affect the resulting distribution. The results for (i) are the content of Figures 1 and 2 (a,b). Figures 3 and 4 show the results for (ii) and figures 5 and 6 show the results for (iii). Figures 3-6 can be found in Appendix D.

Finally, we address the question of why the proposal distribution was built using the particular choice of $\mu$ we made. Indeed one may use `Base Exchange Walk` for *any* homogenous distribution on $2^{[n]}$ to build a sampler, one simply needs to compute the appropriate acceptance probabilities. We restrict our attention to SLC distributions so as to be able to build on the recent mixing time results for homogenous SLC distributions. An obvious alternative to using $\mu$ to build the proposal is to use $\mathcal{H}_d\nu$. Figure 2(c) compares the empirical mixing time of these two chains. The strong empirical improvement justifies our choice of adding the extra rescaling factor $d/e$.

# 7 Discussion

In this paper we introduced strongly log-concave distributions as a promising class of models for diversity. They have flexibility beyond that of strongly Rayleigh distributions, e.g., via exponentiated and cardinality constrained distributions (which do not preserve the SR property). We derived a suite of MCMC samplers for general SLC distributions and associated mixing time bounds. For optimization, we showed that SLC distributions satisfy a weak submodularity property and used this to prove mode finding guarantees.

Still, many open problems remain. Although the mixing time bound has the interesting property of not directly depending on $n$, the $O(2^d)$ dependence seems quite conservative compared to the empirical mixing time results. An important future direction would be to close this gap. More fundamentally, the negative dependence properties of SLC distributions need to be explored in greater detail. Finally, in order for SLC models to be deployed in practice the user needs a way to learn a good SLC model from data, a non-trivial task in general since SLC distribution are non-parametric. However, both exponentiation and cardinality constraint add a single parameter that must be learned. We leave the question of how best to learn these parameters as an important topic for future work.

### Acknowledgements

This work was supported by an NSF-BIGDATA award and the Defense Advanced Research Projects Agency (grant number YFA17 N66001-17-1-4039). The views, opinions, and/or findings contained in this article are those of the author and should not be interpreted as representing the official views or policies, either expressed or implied, of the Defense Advanced Research Projects Agency or the Department of Defense. We thank Matt Staib for helpful comments on the draft.

## Footnotes

[1]This result was independently discovered by Brändén and Huh [10].

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
