[Supplementary Material]

# A Proofs for operations preserving strong log-concavity

In this section we prove Theorems 2, 3, and 4.

## A.1 Closure under scaled homogenization

Let us begin this section by observing that closure under homogenization and symmetric homogenization both fail for strongly log-concave polynomials. The homogenization of a polynomial $f(z) = \sum_{|S| \leq d} c_S z^S$ is $f_{\mathrm{h}}(z, y) = \sum_{|S| \leq d} c_S z^S y^{d-|S|}$, and its symmetric homogenization is $f_{\mathrm{sh}} = \Pi(f_{\mathrm{h}})$.

We will use the following lemma.

**Lemma 13.** *[5] $f(y, z) = a + by + cz + dyz$ with $a, b, c, d \in \mathbb{R}_+$ is SLC if and only if $2bc \geq ad$.*

The counterexample is as follows: by the preceding lemma $f(y, z) = 1 + 2y + z + 3yz$ is SLC. Then note that its homogenization is $f_{\mathrm{h}}(w, y, z) = w^2 + 2wy + wz + 3yz$. A quick computational check then shows that $\nabla^2(f_{\mathrm{h}})$ has eigenvalues $-3.1, 0.4, 4.7$ each to one decimal place. Furthermore the symmetric homogenization of $f$ is,

$$f_{\mathrm{sh}}(w_1, w_2, y, z) = w_1 w_2 + (w_1 + w_2)y + 1/2(w_1 + w_2)z + 3yz,$$

and one may check that $\nabla^2(f_{\mathrm{sh}})$ has eigenvalues $3.8, -3.1, -1.0, 0.3$ to one decimal place. This shows that SLC is not closed under homogenization or symmetric homogenization. So we seek modified operations that are conserved by SLC. In Section 3 we introduced the rescaled homogenization of $f = \sum_{|S| \leq d} c_S z^S$,

$$\mathcal{H}_k f(z, y) = \sum_{|S| \leq k} \frac{c_S}{(k - |S|)!} z^S y^{k-|S|}.$$

**Theorem 14.** *Let $M = ([n], \mathcal{I})$ be a rank $d$ matroid and $f = \sum_{S \in \mathcal{I}} c_S z^S \in \mathbb{R}[z_1, \ldots, z_n]$ be SLC where $c_S > 0$ for all $S \in \mathcal{I}$. For any $k \leq d$ the polynomial $\mathcal{H}_k f$ is SLC.*

A key component of proving this theorem is the following lemma.

**Lemma 15.** *Let $f(z) = \sum_{S \in \mathcal{I}} c_S z^S \in \mathbb{R}[z_1, \ldots, z_n]$ be multiaffine and SLC and suppose that $M = ([n], \mathcal{I})$ is a matroid of rank $d$ and $k \leq d$. Then $\partial_y^{k-2}(\mathcal{H}_k f)$ is log-concave.*

*Proof.* Let $q = \partial_y^{k-2}(\mathcal{H}_k f)$. We compute,

$$q(y, z) = \partial_y^{k-2}\left( \sum_{S \in \mathcal{I}_k} \frac{c_S}{(k - |S|)!} z^S y^{k-|S|} \right)$$

$$= \frac{1}{2} c_{\varnothing} y^2 + y \sum_{\{i\} \in \mathcal{I}_k} c_i z_i + \sum_{\{i,j\} \in \mathcal{I}_k} c_{i,j} z_i z_j$$

Let $Q = \nabla^2 q$. Note that since $q$ is of degree two, $Q$ is in fact a constant and does not depend on $y$ or $z$. Therefore, $q$ is log-concave on $\mathbb{R}_{\geq 0}^{n+1}$ if and only if it is log-concave at $a = (1, 0, \ldots, 0)^\top$. This happens if and only if $(a^\top Q a)Q - (Qa)(Qa)^\top$ is negative semidefinite by Lemma 39. But this is only true if and only if the matrix $[c_{\varnothing} c_{ij} - c_i c_j]_{i,j \in [n]}$ is negative semidefinite. By definition

$$\nabla^2 \log f = \frac{(\nabla^2 f)f - (\nabla f)(\nabla f)^\top}{f^2} \preccurlyeq 0$$

and evaluating at $z = 0$ we notice that $f(0) = c_{\varnothing}$, $\partial_i f(0) = c_i$, and $\partial_{ij} f(0) = c_{ij}$. So indeed we have that $[c_{\varnothing} c_{ij} - c_i c_j]_{i,j \in [n]} \preccurlyeq 0$. □

*Proof of Theorem 14.* To prove strong log-concavity we proceed by verifying the hypotheses of Theorem 36. Let $\alpha \in \mathbb{N}^d$ and $m \in \mathbb{N}$ such that $|\alpha| + m \leq k - 2$. The first order of business is to show that $\partial_z^\alpha \partial_y^m (\mathcal{H}_k f)$ is indecomposable. If $\alpha_i > 1$ for any $i$ then the expression equals $0$, so we may assume $\alpha = \mathbf{1}_K$ for some $K \subseteq [n]$. Then note that

$$
\begin{aligned}
\partial_z^K \partial_y^m (\mathcal{H}_k f) &= \partial_y^m \sum_{S \in \mathcal{I}_k / K} \frac{c_{S \cup K}}{(k - |S| - |K|)!} z^S y^{k - |S| - |K|} \\
&= \partial_y^m \sum_{S \in (\mathcal{I}/K)_{k - |K|}} \frac{c_{S \cup K}}{(k - |S| - |K|)!} z^S y^{k - |S| - |K|} \\
&=: \partial_y^m g,
\end{aligned}
$$

where $\mathcal{I}/K$ is the family of independent sets of $M/K$, the matroid contraction of $M$ by $K$. We first check indecomposability of $\partial_z^K \partial_y^m (\mathcal{H}_k f)$. Note that if $i \in [n] \setminus K$ is a loop of $M/K$ then the variable $z_i$ does not appear in $g$ and $\partial_i g = 0$. Similarly $\partial_i g = 0$ for all $i \in K$. Otherwise the monomial $z_i y^{k-1-m}$ appears in $\partial_y^m g$ with non-zero coefficient. Since $k - 1 - m \geq 1$ this implies that $\partial_i \partial_y g$ is non-zero. In particular the graph formed in the definition of indecomposability is a star centered at $y$ and therefore connected, proving that $\partial_z^K \partial_y^m (\mathcal{H}_k f)$ is indecomposable.

Now suppose that $|K| + m = k - 2$. Notice that $\partial_z^K f = \partial_z^K \sum_{S \in \mathcal{I}} c_S z^S = \sum_{S \in \mathcal{I}/K} c_{S \cup K} z^S$ is SLC, and

$$
\mathcal{H}_{k - |K|} (\partial_z^K f) = \sum_{S \in (\mathcal{I}/K)_{k - |K|}} \frac{c_{S \cup K}}{(k - |S| - |K|)!} z^S y^{k - |S| - |K|}.
$$

So $\partial_z^K \partial_y^m (\mathcal{H}_k f) = \partial_y^m \mathcal{H}_{k - |K|} (\partial_z^K f) = \partial_y^{k - |K| - 2} \mathcal{H}_{k - |K|} (\partial_z^K f)$ and we may apply Lemma 15 to conclude $\partial_z^K \partial_y^m (\mathcal{H}_k f)$ is log-concave. $\qquad\square$

## A.2 Closure under scaled exponentiation

For matrices $A = [a_{ij}]$ and $B = [b_{ij}]$ and scalar $\alpha \in \mathbb{R}$ we write $A^{\circ\alpha}$ to denote the element-wise power $[a_{ij}^\alpha]$ and $A \circ B = [a_{ij} b_{ij}]_{ij}$ to denote the Hadamard (element-wise) product. The proof of Theorem 3 and of Theorem 1.7 from [5] both boil down to the following linear algebra fact.

**Lemma 16.** [2] *Suppose $A = [a_{ij}]$ is symmetric, has non-negative entries and at most one positive eigenvalue. Then $A^{\circ\alpha}$ also has at most one positive eigenvalue for $0 \leq \alpha \leq 1$.*

To prove Lemma 16 we recall a couple of of facts from linear algebra. We shall call a matrix $A$ conditionally negative definite if $z^\top A z \leq 0$ for all $z$ such that $z^\top \mathbf{1} = 0$.

**Lemma 17.** [6] *Suppose $A = [a_{ij}]$ is symmetric, has positive entries, and at most one positive eigenvalue. Then $\left[ \frac{a_{ij}}{v_i v_j} \right]$ is conditionally negative definite, where $v$ is the Perron-Frobenius eigenvector of $A$.*

**Lemma 18.** [7] *Suppose $A \in \mathbb{R}^{n \times n}$ is conditionally negative definite. Then $A^{\circ\alpha}$ is conditionally negative definite for $0 \leq \alpha \leq 1$.*

With these two facts in hand we are now ready to prove Lemma 16.

*Proof of Lemma 16.* Assume that $a_{ij} > 0$ for all $i$ and $j$. The general case is then obtained by a limiting argument. Since $A = [a_{ij}]$ is symmetric, has positive entries, and at most one positive eigenvalue, Lemma 17 implies that $\left[ \frac{a_{ij}}{v_i v_j} \right]$ is conditionally negative definite, where $v$ is the Perron-Frobenius eigenvector of $A$. Then Lemma 18 tell us that

$$B = \left[ \frac{a_{ij}^{\alpha}}{v_i^{\alpha} v_j^{\alpha}} \right] = [a_{ij}^{\alpha}] \circ (vv^{\top})^{\circ-\alpha} = A^{\circ\alpha} \circ (vv^{\top})^{\circ-\alpha}$$

is also conditionally negative definite. Note the identity,

$$A^{\circ\alpha} = [a_{ij}^{\alpha}] = B \circ (vv^{\top})^{\circ\alpha} = \text{diag}(v^{\circ\alpha}) B \text{diag}(v^{\circ\alpha}).$$

Since the entries of the Perron-Frobenius eigenvector are all strictly positive, $\text{diag}(v^{\circ\alpha})$ is non-singular. We may therefore apply Sylvester's law of inertia to conclude that $B$ and $A^{\circ\alpha} = \text{diag}(v^{\circ\alpha}) B \text{diag}(v^{\circ\alpha})$ have the same number of positive eigenvalues: one.

$\square$

Next, we prove Theorem 3 by showing how it reduces to exactly the statement of Lemma 16.

*Proof of Theorem 3.* Consider $S \subseteq [n]$ and $m \in \mathbb{N}$ such that $|S| + m = d - 2$. Denoting $\nabla^2(\partial_z^S \partial_z^m \mathcal{H}_k f) = A = [a_{ij}]_{i,j=1}^{n+1}$ one observes that

$$\nabla^2(\partial_z^S \partial_z^m \mathcal{H}_{k,\alpha} f) = A^{\circ\alpha} = [a_{ij}^{\alpha}]_{i,j=1}^{n+1}.$$

By Theorem 2 the matrix $A$ has at most one positive eigenvalue and so we may apply Lemma 16 to yield the result. $\square$

Note Lemma 18 also permits a simplified proof of the following theorem due to Anari et al. concerning the homogeneous case.

**Theorem 19.** *Suppose $f = \sum_{|S|=d} c_S z^S \in \mathbb{R}_+[z_1, \ldots, z_n]$ is SLC. Then $f_\alpha = \sum_{|S|=d} c_S^\alpha z^S$ is SLC for any $0 \leq \alpha \leq 1$.*

*Proof.* We prove strong log-concavity of $f_\alpha$ by verifying the hypotheses of Theorem 36. Assume $c_S > 0$ for all $|S| = d$. The general case is then obtained by taking point-wise limits. Let $S \subseteq [n]$ be such that $|S| = d - 2$. Then notice that

$$\nabla^2(\partial^T f) = \left[ c_{S \cup \{i,j\}} \right]_{ij} \quad \text{and} \quad \nabla^2(\partial^T f_\alpha) = \left[ c_{S \cup \{i,j\}}^\alpha \right]_{ij}.$$

Since $f$ is SLC, Lemma 39 implies $\nabla^2(\partial^T f)$ has at most one positive eigenvalue. So Lemma 16 implies that $\nabla^2(\partial^T f_\alpha)$ also has at most one positive eigenvalue, which proves the strong log-concavity of $f_\alpha$ by applying Lemma 39 in the other direction. $\square$

It is a reasonable question to ask whether or not the preceding theorem or Theorem 3 can be extended to the regime $\alpha > 1$. This is in fact not the case. First we show that if either holds for *any* $\alpha > 1$ then it must hold for *all* $\alpha > 1$, then we give a counterexample showing that it fails for $\alpha = 2$ for both cases. Note carefully that the conclusion is therefore stronger than a mere existence claim. In fact we may conclude: Theorems 3 and 19 both fail for *all* $\alpha > 1$.

To make the following statement succinct let us define $\mathcal{A}$ to be the set of all symmetric real-valued matrices with non-negative entries and at most one positive eigenvalue.

**Lemma 20.** *Suppose there is a $\alpha^* > 1$ such that: if $A \in \mathcal{A}$ then $A^{\circ\alpha^*} \in \mathcal{A}$. Then for any $\alpha > 1$: if $A \in \mathcal{A}$ then $A^{\circ\alpha} \in \mathcal{A}$*

*Proof.* Let $A \in \mathcal{A}$. We may repeatedly apply the hypothesis to conclude that $A^{\circ m \alpha^*} \in \mathcal{A}$ for *any* $m \in \mathbb{N}$. So in particular we may pick $m$ sufficiently big that $m\alpha^* > \alpha$. But now $\delta = \alpha/m\alpha^* < 1$ so we may apply Lemma 16 to conclude that $A^{\circ\alpha} = A^{\circ\delta m \alpha^*} \in \mathcal{A}$. $\square$

**Corollary 21.** *Suppose there is a $\alpha^* > 1$ such that: if $f = \sum_{|S|=d} c_S z^S \in \mathbb{R}_+[z_1, \ldots, z_n]$ is SLC, then $f_{\alpha^*} = \sum_{|S|=d} c_S^{\alpha^*} z^S$ is SLC. Then for any $\alpha > 1$: if $f \in \mathbb{R}_+[z_1, \ldots, z_n]$ is SLC then $f_\alpha$ is SLC.*

**Corollary 22.** *Suppose there is a $\alpha^* > 1$ such that: if $f = \sum_{|S| \leq d} c_S z^S \in \mathbb{R}_+[z_1, \ldots, z_n]$ is SLC, then $\mathcal{H}_{k,\alpha^*} f$ is SLC for $k \leq d$. Then for any $\alpha > 1$: if $f \in \mathbb{R}_+[z_1, \ldots, z_n]$ is SLC then $\mathcal{H}_{k,\alpha} f$ is SLC.*

For the counterexample, consider $f = 10wx + 3wy + 2wz + 2xy + 6xz$. The Hessian of $f$ equals

$$\nabla^2 f = \begin{bmatrix} 0 & 10 & 3 & 2 \\ 10 & 0 & 2 & 6 \\ 3 & 2 & 0 & 1 \\ 2 & 6 & 1 & 0 \end{bmatrix}.$$

One can numerically check that $\nabla^2 f$ has eigenvalues $\{113.6, -10.9, -2.2, -0.4\}$ to one decimal place, so $f$ is log-concave by Lemma 39 and hence SLC since it is of degree 2. However, $f_2$ has Hessian equal to

$$\nabla^2 f_2 = \begin{bmatrix} 0 & 100 & 9 & 4 \\ 100 & 0 & 4 & 36 \\ 9 & 4 & 0 & 1 \\ 4 & 36 & 1 & 0 \end{bmatrix}.$$

which has eigenvalues $\{108.4, -105.2, -4.0, 0.8\}$ to one decimal place. So $f_2$ is not SLC.

The same example can be used to build a counterexample to Theorem 3 in the regime $\alpha > 1$. Indeed setting $w = 1$ in $f$ we obtain an SLC polynomial $g = 10x + 3y + 2z + 2xy + 6xz$ such that $\mathcal{H}_{2,2} g = f_2$ is not SLC.

### A.3  Closure under polarization

We begin by observing an algebraic identity that allows one to push derivatives inside the polarization operation $\Pi$.

**Lemma 23.** *Let $f = \sum_{|S| \leq d} c_S z^S y^{d-|S|} \in \mathbb{R}[z_1, \ldots, z_n, y]$. Then $\partial_{y_i} \Pi(f) = \frac{1}{d} \Pi(\partial_y f)$ and $\partial_{z_j} \Pi(f) = \Pi(\partial_{z_j} f)$ for $i \in [d]$ and $j \in [n]$.*

*Proof.* Since $\Pi(f) \in \mathbb{R}[z_1, \ldots, z_n, y_1, \ldots, y_d]$ is symmetric in $y_1, \ldots, y_d$, to prove the $y_i$ part of the claim it suffices to prove the claim for $\partial_{y_d} \Pi(f)$ only. Recall that the polarization of $f$ is,

$$\Pi(f)(z_1, \ldots, z_n, y_1, \ldots, y_d) = \sum_{|S| \leq d} c_S z^S \binom{d}{|S|}^{-1} e_{d-|S|}(y_1, \ldots, y_d)$$

where $e_k(y_1, \ldots, y_d)$ is the $k$th elementary symmetric polynomial in $d$ variables. We begin computing directly,

$$\partial_{y_d}\Pi(f) = \partial_{y_d}\left(\sum_{|S|\leq d} c_S z^S \binom{d}{|S|}^{-1} e_{d-|S|}(y_1,\ldots,y_d)\right)$$

$$= \sum_{|S|\leq d} c_S z^S \binom{d}{|S|}^{-1} \partial_{y_d}\left(e_{d-|S|}(y_1,\ldots,y_d)\right)$$

$$= \sum_{|S|<d} c_S z^S \binom{d}{|S|}^{-1} e_{d-|S|-1}(y_1,\ldots,y_{d-1}))$$

$$= \sum_{|S|<d} c_S z^S \left(\frac{1}{d}(d-|S|)\binom{d-1}{|S|}^{-1}\right) e_{d-|S|-1}(y_1,\ldots,y_{d-1})$$

$$= \frac{1}{d}\sum_{|S|<d} (d-|S|)c_S z^S \binom{d-1}{|S|}^{-1} e_{d-|S|-1}(y_1,\ldots,y_{d-1})$$

$$= \frac{1}{d}\Pi\left(\sum_{|S|<d}(d-|S|)c_S z^S y^{d-1-|S|}\right)$$

$$= \frac{1}{d}\Pi\left(\partial_y\left\{\sum_{|S|\leq d} c_S z^S y^{d-|S|}\right\}\right)$$

$$= \frac{1}{d}\Pi(\partial_y f),$$

where we used the elementary relation

$$\binom{d}{|S|}^{-1} = \frac{1}{d}(d-|S|)\binom{d-1}{|S|}^{-1}.$$

The $\partial_{z_j}\Pi(f)$ part of the claim follows by a similar, but simpler, calculation. $\qquad\square$

**Corollary 24.** *Let* $f = \sum_{|S|\leq d} c_S z^S y^{d-|S|} \in \mathbb{R}[z_1,\ldots,z_n,y]$. *Then* $\partial_z^\alpha \partial_y^\beta \Pi(f) = c\Pi(\partial_z^\alpha \partial_y^{|\beta|} f)$ *for any* $\alpha, \beta \in \mathbb{N}^n$ *where* $c > 0$ *is some constant.*

**Lemma 25.** $f = \sum_{|S|\leq d} c_S z^S y^{d-|S|}$ *is indecomposable if and only if* $\Pi(f)$ *is indecomposable.*

*Proof.* $f = 0$ if and only if $\Pi(f) = 0$ and if this is the case then we are done. So suppose $f$ is not identically 0. Similarly if $c_S = 0$ for all $|S| < d$ we are done since $f = \Pi(f)$. Let $\mathcal{Z} = \{z_1,\ldots,z_n\}$ and $\mathcal{Y} = \{y_1,\ldots,y_d\}$. Suppose that $\Pi(f)$ is *not* indecomposable. Then there exists partitions $\mathcal{Z}_1 \cup \mathcal{Z}_2 = \mathcal{Z}$ and $\mathcal{Y}_1 \cup \mathcal{Y}_2 = \mathcal{Y}$ such that $\Pi(f) = g_1 + g_1$ where neither $g_1$ nor $g_2$ are identically 0 and $g_1$ depends only on the variables $\mathcal{Z}_1 \cup \mathcal{Y}_1$ and $g_2$ depends only on the variables $\mathcal{Z}_2 \cup \mathcal{Y}_2$. However we must have $\{\mathcal{Y}_1, \mathcal{Y}_2\} = \{\varnothing, \mathcal{Y}\}$ since we may pick an $S$ with $|S| < d$ such that $c_S \neq 0$, and so $\Pi(f)$ contains the sum of monomials $c_S z^S \binom{d}{|S|}^{-1} e_{d-|S|}(y_1,\ldots,y_d)$, which includes some terms containing $y_i$ and $y_j$ for any distinct $i,j \in [d]$. This observation is sufficient since all coefficients are non-negative and hence do not cancel each other out. Without loss of generality suppose $\mathcal{Y}_1 = \mathcal{Y}$ and $\mathcal{Y}_1 = \varnothing$. However, upon setting $y = y_1 = \ldots = y_d$ in $g_1$ we discover that we may write $f = g_1 + g_2$ where $g_1$ depends only on $\mathcal{Z}_1 \cup \{y\}$ and $g_2$ depends only on $\mathcal{Z}_2$. In other words, $f$ is not indecomposable.

Conversely, suppose $f$ is not indecomposable. Then we may write $f = g_1 + g_2$ where $g_1$ depends only on $\mathcal{Z}_1 \cup \{y\}$ and $g_2$ depends only on $\mathcal{Z}_2$ were $\mathcal{Z} = \mathcal{Z}_1 \cup \mathcal{Z}_2$ is a partition. But, since polarization is a linear operator, this implies that $\Pi(f)$ may be decomposed into the sum of two non-zero polynomials, one depending only on $\mathcal{Z}_1 \cup \mathcal{Y}$ and the other on $\mathcal{Z}_2$. So $\Pi(f)$ is not indecomposable either. $\qquad\square$

We recall a fact from linear algebra that we shall use in the coming proof.

**Lemma 26.** *Suppose $A \in \mathbb{R}^{n \times n}$ is symmetric and let $R \in \mathbb{R}^{m \times n}$. If $A$ has at most one positive eigenvalue, then $RAR^\top$ has at most one positive eigenvalue.*

*Proof of Theorem 4.* Suppose $\Pi(f)$ is SLC. Then by definition, upon setting $y_i = y$ for all $i$ we obtain $f$. Therefore $f$ is SLC since Lemma 38 states that the SLC property is invariant under affine transformations of the coordinates. Conversely, suppose $f$ is SLC. We shall check that $\Pi(f)$ satisfies the hypotheses of Theorem 36. Note that $d = \deg(f) = \deg(\Pi(f))$. Take any $\alpha, \beta \in \mathbb{N}^n$ such that $|\alpha| + |\beta| \leq d - 2$. Then by Corollary 24, $\partial_z^\alpha \partial_y^\beta \Pi(f) = c\Pi(\partial_z^\alpha \partial_y^{|\beta|} f)$ for some constant $c$. Since $f$ is SLC, so is $\partial_z^\alpha \partial_y^{|\beta|} f$ and so by Theorem 40 its support is M-convex (see Theorem 40 for definintion), which implies indecomposability. Hence by Lemma 25 $\partial_z^\alpha \partial_y^\beta \Pi(f)$ is indecomposable too. Now suppose that $|\alpha| + |\beta| = d - 2$. We must verify that $\partial_z^\alpha \partial_y^\beta \Pi(f) = c\Pi(\partial_z^\alpha \partial_y^{|\beta|} f)$ is log-concave. $g := \partial_z^\alpha \partial_y^{|\beta|} f$ is homogeneous, of degree 2, and multiaffine in all except one coordinate. Hence $g$ is of the form

$$g = \sum_{i,j=1}^n c_{ij} z_i z_j + y \sum_{i=1}^n c_i z_i + c_\varnothing y^2.$$

Since $f$ is SLC, $g$ is log-concave. By Lemma 39 this implies that

$$\nabla^2 g = \begin{bmatrix} c_\varnothing & \cdots & c_j & \cdots \\ \vdots & & \vdots & \\ c_i & \cdots & c_{ij} & \cdots \\ \vdots & & \vdots & \end{bmatrix}$$

(a constant) has at most one positive eigenvalue. We can explicitly write

$$\Pi(g) = \sum_{i,j=1}^n c_{ij} z_i z_j + \frac{1}{2}(y_1 + y_2) \sum_{i=1}^n c_i z_i + c_\varnothing y_1 y_2.$$

and therefore

$$\nabla^2\big(\Pi(g)\big) = \begin{bmatrix} 0 & c_\varnothing & \cdots & \frac{1}{2}c_j & \cdots \\ c_\varnothing & 0 & \cdots & \frac{1}{2}c_j & \cdots \\ \vdots & \vdots & & \vdots & \\ \frac{1}{2}c_i & \frac{1}{2}c_i & \cdots & c_{ij} & \cdots \\ \vdots & \vdots & & \vdots & \end{bmatrix}.$$

It suffices now to show that $\nabla^2\big(\Pi(g)\big)$ also has at most one positive eigenvalue. Consider the $(n+1) \times n$ matrix

$$R = \begin{bmatrix} 1/2 & & & \\ 1/2 & 1 & & \\ & & \ddots & \\ & & & 1 \end{bmatrix}$$

where the entries along the specified diagonal are ones, and everywhere else is zeros. By Lemma 26, the following $(n+1) \times (n+1)$ matrix also has at most one positive eigenvalue,

$$R(\nabla^2 g)R^\top = \begin{bmatrix} \frac{1}{2}c_\varnothing & \frac{1}{2}c_\varnothing & \cdots & \frac{1}{2}c_j & \cdots \\ \frac{1}{2}c_\varnothing & \frac{1}{2}c_\varnothing & \cdots & \frac{1}{2}c_j & \cdots \\ \vdots & \vdots & & \vdots & \\ \frac{1}{2}c_i & \frac{1}{2}c_i & \cdots & c_{ij} & \cdots \\ \vdots & \vdots & & \vdots & \end{bmatrix}.$$

Finally observe that $\nabla^2\big(\Pi(g)\big) + uu^\top = R(\nabla^2 g)R^\top$ where $u = (\sqrt{c_\varnothing/2}, -\sqrt{c_\varnothing/2}, 0, \ldots, 0)^\top$. Cauchy's Interlacing Theorem implies that $\nabla^2\big(\Pi(g)\big)$ has at most one positive eigenvalue. $\square$

### A.4  Closure under constraining to subsets of a given size

We state two propositions which when translated into probabilistic language say that if $\pi$ is SLC then the distribution proportional to $\pi(S)\mathbf{1}\{|S| = k\}$ is also SLC.

**Lemma 27.** *Let $f = \sum_{|\alpha| \le d} c_\alpha z^\alpha \in \mathbb{R}_+[z_1, \ldots, z_n]$ be of degree $d$ and be SLC. Then $f_d(z) = \sum_{|\alpha|=d} c_\alpha z^\alpha$ is also SLC.*

**Corollary 28.** *Let $f = \sum_{|\alpha| \le d} c_\alpha z^\alpha \in \mathbb{R}_+[z_1, \ldots, z_n]$ be of degree $d$ and be SLC. Then for any $k \le d$, $f_k(z) = \sum_{|\alpha|=k} c_\alpha z^\alpha$ is also SLC.*

*Proof of Lemma 27.* Assume $c_\alpha > 0$ for all $|\alpha| \le d$. The general result is then obtained by taking point-wise limits of coefficients. So for any $|\alpha| \le d-2$, the polynomial $\partial^\alpha f_d$ is evidently inde-composable. Now assume $|\alpha| = d-2$. Then note that $\nabla^2(\partial^\alpha f) = \nabla^2(\partial^\alpha f_d)$. Since $f$ is strongly log-concave $\nabla^2(\partial^\alpha f)$ has at most one positive eigenvalue, and hence so does $\nabla^2(\partial^\alpha f_d)$. $\square$

*Proof of Corollary 28.* This result immediately follows from combining Theorem 2 and Lemma 27. $\square$

## B  The Metropolis-Hastings chain's stationary distribution and mixing time

### B.1  The chain has the right stationary distribution

We only need to check that the acceptance probability stated does indeed yield a chain with stationary distribution $\nu_{\text{sh}}$. We build our algorithm using the usual Metropolis-Hastings procedure. We consider the proposal distributions $Q$, each being Anari's base exchange kernel for the distribution $\mu$. The Metropolis-Hastings algorithm is as follows,

1. Suppose the current state is $S \subseteq [n+d]$ with $|S \cap [n]| = k$,
2. Sample $T \sim Q(S, \cdot)$,
3. Compute the acceptance probability $a = \min\left\{1, \frac{\nu_{\text{sh}}(T)}{\nu_{\text{sh}}(S)} \frac{Q(T,S)}{Q(S,T)}\right\}$,
4. With probability $a$, update $S \leftarrow T$,
5. Otherwise do not update.

All that remains is to compute the acceptance probability. If $S = T$ then clearly $a = 1$ and Algorithm 1 is in agreement. So suppose from now on that $S \ne T$. If further we have $|S \cap T| < d-1$ then $Q(S,T) = 0$ and so we may discount this possibility since the proposal distribution will never sample such a $T$. This leaves only the case that $|S \cap T| = d-1$. We may write the transition kernel explicitly

$$Q(S,T) = \frac{1}{d}\frac{\mu(T)}{w(S \cap T)} \propto \frac{1}{d}\left(\frac{d}{e}\right)^{d-|T \cap [n]|}\frac{\nu_{\text{swh}}(T)}{w(S \cap T)}$$

where we define $w(S \cap T) = \sum_{i \in (S \cap T)^c} \mu((S \cap T) \cup i)$. Computing the ratio

$$\frac{Q(T,S)}{Q(S,T)} = \frac{1/d \cdot \mu(S)/w(S \cap T)}{1/d \cdot \mu(T)/w(S \cap T)}$$

$$= \frac{\mu(S)}{\mu(T)}$$

$$= \frac{(d - |T \cap [n]|)!}{(d - |S \cap [n]|)!} \frac{(d/e)^{d - |S \cap [n]|}}{(d/e)^{d - |T \cap [n]|}} \frac{\nu_{\mathrm{sh}}(S)}{\nu_{\mathrm{sh}}(T)}$$

$$= \left(\frac{d}{e}\right)^{|T \cap [n]| - k} \frac{(d - |T \cap [n]|)!}{(d - k)!} \frac{\nu_{\mathrm{sh}}(S)}{\nu_{\mathrm{sh}}(T)},$$

and so,

$$a = \min\left\{1, \frac{\nu_{\mathrm{sh}}(T)}{\nu_{\mathrm{sh}}(S)} \left(\frac{d}{e}\right)^{|T \cap [n]| - k} \frac{(d - |T \cap [n]|)!}{(d - k)!} \frac{\nu_{\mathrm{sh}}(S)}{\nu_{\mathrm{sh}}(T)}\right\}$$

$$= \min\left\{1, \left(\frac{d}{e}\right)^{|T \cap [n]| - k} \frac{(d - |T \cap [n]|)!}{(d - k)!}\right\}.$$

This leaves the following three cases,

1. If $|T \cap [n]| = k - 1$, then $a = \min\{1, \frac{e}{d}(d - k + 1)\}$.
2. If $|T \cap [n]| = k$, then $a = \min\{1, 1\} = 1$.
3. If $|T \cap [n]| = k + 1$, then $a = \min\{1, \frac{d}{e} \frac{1}{(d-k)}\}$.

These are exactly the acceptance probabilities given in Algorithm 1.

## B.2 Mixing time bounds

The mixing time of a chain can be bounded using an important quantity known as the log-Sobolev constant. A famous theorem due to Diaconis and Saloff-Coste [19] shows that if $(Q, \pi)$ has log-Sobolev constant $\alpha$, then the mixing time of this chain is bounded by $t_{S_0}(\varepsilon) \leq \frac{1}{\alpha}\left(\log\log \frac{1}{\pi(S_0)} + \log \frac{1}{2\varepsilon^2}\right)$. For a particular problem instance the objective is therefore to find a lower bound on $\alpha$.

Let $P$ denote the transition kernel described in Algorithm 1 that we have just confirmed have stationary distribution $\nu_{\mathrm{sh}}$. In this section we shall prove Theorem 8, a mixing time bound on the chain $(P, \nu_{\mathrm{sh}})$. The bound is obtained by combining two pieces of information: the fact, obtained by Cryan [15], that the chain $(Q, \nu_{\mathrm{swh}})$ has log-Sobolev constant bounded below by $1/d$, and a theorem due to Diaconis and Saloff-Coste [19] that allows one to compare the log-Sobolev constants of two chains. For the statement of the comparison theorem see appendix E.2.

Recall that $P$ is constructed using the base exchange walk for the following distribution

$$\mu(S) = \frac{1}{Z} \frac{1}{(d - |S \cap [n]|)!} \left(\frac{d}{e}\right)^{d - |S \cap [n]|} \nu_{\mathrm{sh}}(S)$$

for $S \subseteq [n + d]$ where $Z$ is the partition function of $\mu$.

**Lemma 29.**

$$\frac{\sqrt{2\pi}}{2^d} Z \leq \frac{\nu_{sh}(S)}{\mu(S)} \leq e\sqrt{d}Z$$

for all $S \subseteq [n + d]$.

*Proof.* For this proof set $k = |S \cap [n]|$. Then

$$\frac{1}{Z} \frac{\nu_{\text{sh}}(S)}{\mu(S)} = (d-k)! \left(\frac{e}{d}\right)^{d-k}.$$

Stirling's approximation says that

$$\sqrt{2\pi}\sqrt{d-k}\left(\frac{d-k}{e}\right)^{d-k} \leq (d-k)! \leq e\sqrt{d-k}\left(\frac{d-k}{e}\right)^{d-k}$$

which upon substitution into (B.2) yields

$$\sqrt{2\pi}\sqrt{d-k}\left(\frac{d-k}{d}\right)^{d-k} \leq (d-k)! \left(\frac{e}{d}\right)^{d-k} \leq e\sqrt{d-k}\left(\frac{d-k}{d}\right)^{d-k}.$$

Inserting $k = 0$ one observes that $e\sqrt{d}$ is a tight upper bound. It remains only to find a lower bound. Note that if $k = d$ then $\frac{1}{Z}\frac{\nu_{\text{sh}}(S)}{\mu(S)} = 1$. So we may assume that $k \in [d-1]$. We may bound below as follows,

$$
\begin{aligned}
\sqrt{2\pi}\sqrt{d-k}\left(\frac{d-k}{d}\right)^{d-k} &\geq \sqrt{2\pi}\left(\frac{d-k}{d}\right)^{d-k} \\
&= \sqrt{2\pi}\left(\frac{d}{d-k}\right)^{-(d-k)} \\
&= \sqrt{2\pi}\left(1 + \frac{k}{d-k}\right)^{-(d-k)} \\
&\geq \min_{k_1,k_2 \in [d-1]} \sqrt{2\pi}\left(1 + \frac{k_1}{d-k_2}\right)^{-(d-k_2)} \\
&\geq \min_{k_1 \in [d-1]} \sqrt{2\pi}\left(1 + \frac{k_1}{d}\right)^{-d} \\
&\geq \sqrt{2\pi}\left(1 + \frac{d}{d}\right)^{-d} \\
&= \sqrt{2\pi}\,2^{-d}.
\end{aligned}
$$

where the penultimate inquality is due to the fact that $(1 + x/n)^n$ is an increasing function of $n$ for any fixed $x \in \mathbb{R}$. $\square$

*Proof of Theorem 8.* We seek $a, A > 0$ like in the statement of Theorem 42. Lemma 29 says exactly that for all $S \subseteq [n+d]$,

$$\frac{\sqrt{2\pi}}{2^d} Z \leq \frac{\nu_{\text{sh}}(S)}{\mu(S)} \leq e\sqrt{d}Z$$

where here $Z$ is the partition function of $\mu$. So we may take $a = e\sqrt{d}Z$. Next, recall that $P$ denotes the transition kernel defined by Algorithm 1. We may write $P$ explicitly

$$
P(S,T) = \begin{cases}
a(S,T)Q(S,T) & \text{if } |T| = k-1 \\
Q(S,T) & \text{if } |T| = k \text{ and } T \neq S \\
Q(S,T) + \sum_{|R|=k\pm 1} (1 - a(S,R))Q(S,R) & \text{if } T = S \\
a(S,T)Q(S,T) & \text{if } |T| = k+1,
\end{cases}
$$

where $a(S, T)$ denotes the previously defined acceptance probability for a proposed move to $T$ if the current state is $S$. Observe that one divided by the acceptance probability is bounded above by $\max\{e, d/e\} = d/e$ for $d \geq 8$. So we see that $Q \leq \frac{d}{e}P$. Combining this with the lower bound on $\nu_{\text{sh}}(S)/\mu(S)$ we find that we may take

$$A = \frac{d}{e}\frac{2^d}{\sqrt{2\pi Z}}.$$

Applying Theorem 42 and using the fact that the log-Sobolev constant of $(Q, \nu_{\text{swh}})$ is bounded below by $1/d$ we obtain the following lower bound on the log-Sobolev constant for $(P, \nu_{\text{sh}})$,

$$\frac{1}{aA}\frac{1}{d} = \frac{e\sqrt{2\pi}}{d^{5/2}2^d}.$$

$\square$

## C   Proof of weak log-submodular properties

In this section we prove Theorems 10, 11, Corollary 12 and give two other simple greedy algorithms for non-negative weakly log-submodular functions and monotone weakly log-submodular functions respectively. Whilst we only derive results for three algorithms here, we expect that many submodular maximization algorithms will have weak submodular analogues, including guarantees.

In this section we shall use the notation $\nu(e \mid S) = \nu(S \cup \{e\}) - \nu(S)$ for any set function $\nu : 2^{[n]} \to \mathbb{R}_+$ and any $S \subseteq [n]$ and $i \in [n]$.

### C.1   Functions whose weighted homogenized generating polynomial is SLC are weakly log-submodular

*Proof of Theorem 10.* By Theorem 2, the polynomial $g = \mathcal{H}_n f$ is SLC where $f$ is the generating polynomial of $\nu$. Hence so is $\partial_z^S g$ for any $S \subseteq [n]$. Applying Theorem 37 with $z = 0$ and $y = 1$ we obtain,

$$\partial_z^S g(0,1)\partial_i\partial_j\partial_z^S g(0,1) \leq 2\left(1 - \frac{1}{d}\right)\partial_i\partial_z^S g(0,1)\partial_j\partial_z^S g(0,1).$$

Rewriting this in terms of $\nu$ gives,

$$\frac{\nu(S)}{(n-|S|)!}\frac{\nu(S \cup \{i,j\})}{(n-|S|-2)!} \leq 2\left(1 - \frac{1}{d}\right)\frac{\nu(S \cup i)}{(n-|S|-1)!}\frac{\nu(S \cup j)}{(n-|S|-1)!},$$

which rearranges to,

$$\nu(S)\nu(S \cup \{i,j\}) \leq 2\left(1 - \frac{1}{d}\right)\left(\frac{n-|S|}{n-|S|-1}\right)\nu(S \cup i)\nu(S \cup j).$$

Finally note that $\frac{n-|S|}{n-|S|-1} \leq 2$ since $n - |S| \geq 2$. $\square$

### C.2   Distorted greedy guarantees

We include results from [31] in the interests of completeness. Recall the definition $\Phi_i(S) = (1 - 1/k)^{k-i}\eta(S) - c(S)$. We also introduce the following object that will be useful for our analysis,

$$\Psi_i(S, e) = \max\left\{0, \left(1 - \frac{1}{k}\right)^{k-(i+1)}\eta(e \mid S) - c_e\right\}$$

for $i = 0, \ldots k - 1$. Finally, note that by writing $\nu(\text{OPT}) - \nu(S)$ as a telescoping sum and using the definition of weak submodularity one finds that,

$$\nu(\text{OPT}) - \nu(S) \leq \sum_{e \in \text{OPT}} \nu(e \mid S) + \frac{1}{2}\ell(\ell - 1)\gamma.$$

This fact will be used in the proof of Lemma 31. We prepare for the proof of Theorem 11 by recalling two lemmas.

**Lemma 30.** *[31] In each iteration of the distorted greedy Algorithm 2 we have,*

$$\Phi_{i+1}(S_{i+1}) - \Phi_i(S_i) = \Psi_i(S_i, e_i) + \frac{1}{k}\left(1 - \frac{1}{k}\right)^{k-(i+1)}\eta(S_i).$$

*Proof.* See [31]. □

**Lemma 31.** *In each iteration of the distorted greedy Algorithm 2 we have,*

$$\Psi_i(S_i, e_i) \geq \frac{1}{k}\left(1 - \frac{1}{k}\right)^{k-(i+1)}\left(\eta(OPT) - \eta(S_i) - \frac{1}{2}\ell(\ell - 1)\gamma\right) - \frac{1}{k}c(OPT)$$

*Proof of 31 (adapted from [31]).*

$$k \cdot \Psi_i(S_i, e_i) = k \max_{e \in [n]} \left\{0, \left(1 - 1/k\right)^{k-(i+1)}\eta(e \mid S) - c_e\right\} \tag{2}$$

$$\geq |\text{OPT}| \max_{e \in \text{OPT}} \left\{0, \left(1 - 1/k\right)^{k-(i+1)}\eta(e \mid S) - c_e\right\} \tag{3}$$

$$\geq \sum_{e \in \text{OPT}} \left(\left(1 - 1/k\right)^{k-(i+1)}\eta(e \mid S) - c_e\right) \tag{4}$$

$$= \left(1 - 1/k\right)^{k-(i+1)} \sum_{e \in \text{OPT}} \eta(e \mid S) - c(\text{OPT}) \tag{5}$$

$$\geq \left(1 - 1/k\right)^{k-(i+1)}\left(\eta(\text{OPT}) - \eta(S_i) - \frac{1}{2}\ell(\ell - 1)\gamma\right) - c(\text{OPT}). \tag{6}$$

Inequality (3) follows since $|\text{OPT}| \leq k$ and restricting the domain of the maximum can only make the expression smaller, (4) follows since the average of a collection of numbers is no bigger than the maximum, and (6) follows from the $\gamma$-weak submodularity of $\nu$. □

*Proof of Theorem 11 (adapted from [31]).* Observe that

$$\Phi_0(S_0) = \left(1 - \frac{1}{k}\right)^k \eta(\varnothing) - c(\varnothing) = 0,$$

and

$$\Phi_k(S_k) = \left(1 - \frac{1}{k}\right)^0 \eta(R) - c(R) = \eta(R) - c(R).$$

Now,

$$\nu(R) = \eta(R) - c(R) = \Phi_k(S_k) - \Phi_0(S_0)$$

$$= \sum_{i=0}^{k-1}\left(\Phi_{i+1}(S_{i+1}) - \Phi_i(S_i)\right). \tag{7}$$

Applying Lemma 30 then 31 we find,

$$
\begin{aligned}
\Phi_{i+1}(S_{i+1}) - \Phi_i(S_i) &= \Psi_i(S_i, e_i) + \frac{1}{k}\left(1 - \frac{1}{k}\right)^{k-(i+1)} \eta(S_i) \\
&\geq \frac{1}{k}\left(1 - \frac{1}{k}\right)^{k-(i+1)} \left(\eta(\text{OPT}) - \eta(S_i) - \frac{1}{2}\ell(\ell-1)\gamma\right) - \frac{1}{k}c(\text{OPT}) \\
&\quad + \frac{1}{k}\left(1 - \frac{1}{k}\right)^{k-(i+1)} \eta(S_i) \\
&\geq \frac{1}{k}\left(1 - \frac{1}{k}\right)^{k-(i+1)} \left(\eta(\text{OPT}) - \frac{1}{2}\ell(\ell-1)\gamma\right) - \frac{1}{k}c(\text{OPT})
\end{aligned}
$$

Which, upon inserting into equation (7) yields,

$$
\begin{aligned}
\nu(R) &= \sum_{i=0}^{k-1} \left\{ \frac{1}{k}\left(1 - \frac{1}{k}\right)^{k-(i+1)} \left(\eta(\text{OPT}) - \frac{1}{2}\ell(\ell-1)\gamma\right) - \frac{1}{k}c(\text{OPT}) \right\} \\
&= \left(1 - (1 - \frac{1}{k})^k\right)\left(\eta(\text{OPT}) - \frac{1}{2}\ell(\ell-1)\gamma\right) - c(\text{OPT}) \\
&\geq (1 - 1/e)\left(\eta(\text{OPT}) - \frac{1}{2}\ell(\ell-1)\gamma\right) - c(\text{OPT})s.
\end{aligned}
$$

$\square$

### C.3 Double greedy guarantees

In this section we introduce a modified double greedy unconstrained maximization algorithm and give theoretical guarantees in the case that $\nu$ is weakly submodular and non-negative.

---

**Algorithm 3** Double greedy weak submodular maximization of $\nu$

1: Let $X_0 = \varnothing$ and $Y_0 = [n]$
2: **for** $i = 1, \ldots, n$ **do**
3:     Set $a_i = \max\{\nu(X_{i-1} \cup i) - \nu(X_{i-1}) + (n-i)\gamma, 0\}$
4:     Set $b_i = \max\{\nu(Y_{i-1} \setminus i) - \nu(Y_{i-1}) + (n-i)\gamma, 0\}$
5:     Sample $u \in [0,1]$ uniformly
6:     **if** $u < a_i/(a_i + b_i)$ **then** $X_i \leftarrow X_{i-1} \cup i$, and $Y_i \leftarrow Y_{i-1}$
7:     **else** $X_i \leftarrow X_{i-1}$, and $Y_i \leftarrow Y_{i-1} \setminus i$
8: **return** $X_n = Y_n$

---

Next we give a double greedy unconstrained optimization algorithm. We shall use the convention that $a_i/(a_i + b_i) = 0$ whenever $a_i = b_i = 0$. We denote an element of the set of maximizers $\arg\max_S \nu(S)$ by OPT. It is clear that Algorithm 3 runs in linear time in $n$. It will be useful to notice that by repeatedly applying the definition of weak submodularity one finds that $\nu(T \cup i) - \nu(T) \leq |T \setminus S|\gamma + \nu(S \cup i) - \nu(S)$ for any $S \subseteq T$.

**Theorem 32.** *Suppose $\nu : 2^{[n]} \to \mathbb{R}_+$ is $\gamma$-weakly submodular. Then the solution obtained by the double greedy algorithm $X_n = Y_n$ satisfies*

$$
\mathbb{E}[\nu(X_n)] \geq \frac{1}{2}\nu(OPT) - \frac{3}{16}n(n-1)\gamma = \frac{1}{2}\nu(OPT) - O(n^2).
$$

The following lemma contains the bulk of the work required to prove Theorem 32. The proof is a generalization to the weak submodular setting of the proof of the original double greedy $1/2$-approximation theorem [12] for submodular functions.

**Lemma 33.** *Let $\nu : 2^{[n]} \to \mathbb{R}_+$ be $\gamma$-weakly submodular. Then for $i = 1, \ldots, n$ we have,*

$$\mathbb{E}[\nu(OPT_{i-1}) - \nu(OPT_i)] \leq \frac{1}{2}\mathbb{E}[\nu(X_i) - \nu(X_{i-1}) + \nu(Y_i) - \nu(Y_{i-1})] + \frac{3}{4}(n-i)\gamma$$

*Proof.* It suffices to prove the claim conditioning on the event that $X_{i-1} = A_{i-1}$ for any $A_{i-1}$ such that $X_{i-1} = A_{i-1}$ has non-zero probability. In particular therefore $A_{i-1} \subseteq [i-1]$. Fixing such an event we shall implicitly assume we have conditioned on this event for the rest of the proof. This means that the variables $X_{i-1}, Y_{i-1}, \mathrm{OPT}_{i-1}, a_i$, and $b_i$ all become deterministic. First note that

$$
\begin{aligned}
\mathbb{E}[\nu(X_i) - \nu(X_{i-1}) &+ \nu(Y_i) - \nu(Y_{i-1}))] + (n-i)\gamma/2 \\
&= \frac{a_i}{a_i + b_i}\Big(\nu(X_{i-1} \cup i) - \nu(X_{i-1}) + (n-i)\gamma/2\Big) \\
&\quad + \frac{b_i}{a_i + b_i}\Big(\nu(Y_{i-1} \setminus i) - \nu(Y_{i-1}) + (n-i)\gamma/2\Big) \\
&= \frac{a_i^2 + b_i^2}{a_i + b_i}
\end{aligned}
$$

where the first equality follows by definition of the expectation. To see why the second equality holds first recall that due to $\gamma$-weak submodularity of $\nu$ we have,

$$\nu(X_{i-1} \cup i) - \nu(X_{i-1}) + \nu(Y_{i-1} \setminus i) - \nu(Y_{i-1}) + (n-i)\gamma \geq 0,$$

where the coefficient in front of $\gamma$ is obtained by noting that $\big|(Y_{i-1} \setminus i) \setminus X_{i-1}\big| = \big|(Y_{i-1} \setminus X_{i-1} \cup i\big| = n - i$. Now suppose that $\nu(X_{i-1} \cup i) - \nu(X_{i-1}) + (n-i)\gamma/2 < 0$. Then equation C.3 implies that we must have $\nu(Y_{i-1} \setminus i) - \nu(Y_{i-1}) + (n-i)\gamma/2 > 0$. This implies that $a_i = 0$, $b_i > 0$ and the claimed equality is then readily seen to hold. Alternatively, if $\nu(Y_{i-1} \setminus i) - \nu(Y_{i-1}) + (n-i)\gamma/2 < 0$. Then $\nu(X_{i-1} \cup i) - \nu(X_{i-1}) + (n-i)\gamma/2 > 0$. This implies that $a_i > 0$, $b_i = 0$ and once again the claimed equality holds. The remaining case, where both expressions are non-negative, is the simple case and also holds.

Now let us bound the left hand side expression. We consider two cases. First suppose that $i \notin \mathrm{OPT}$. Then $\mathrm{OPT}_i = \mathrm{OPT}_{i-1} \cup i$ if the updates were $X_i \leftarrow X_{i-1} \cup i$, and $Y_i \leftarrow Y_{i-1}$ and $\mathrm{OPT}_i = \mathrm{OPT}_{i-1}$ if the updates were $X_i \leftarrow X_{i-1}$, and $Y_i \leftarrow Y_{i-1} \setminus i$. Hence,

$$
\begin{aligned}
\mathbb{E}[\nu(\mathrm{OPT}_{i-1}) - \nu(\mathrm{OPT}_i)] &= \frac{a_i}{a_i + b_i}[\nu(\mathrm{OPT}_{i-1}) - \nu(\mathrm{OPT}_{i-1} \cup i)] \\
&\leq \frac{a_i}{a_i + b_i}[\nu(Y_{i-1} \setminus i) - \nu(Y_{i-1}) + (n-i)\gamma] \\
&\leq \frac{a_i b_i}{a_i + b_i} + (n-i)\gamma/2.
\end{aligned}
$$

The first inequality follows due the weak submodularity and the fact that $(Y_{i-1} \setminus i) \setminus \mathrm{OPT}_{i-1} = Y_{i-1} \setminus (\mathrm{OPT}_{i-1} \cup i) \subseteq Y_{i-1} \setminus (X_{i-1} \cup i)$ which has cardinality $n - i$. The second simply follows from the fact that $\nu(Y_{i-1} \setminus i) - \nu(Y_{i-1}) + (n-i)\gamma/2 \leq b_i$ and $a_i/(a_i + b_i) \leq 1$.

Alternatively, suppose $i \in \mathrm{OPT}$. Then $\mathrm{OPT}_i = \mathrm{OPT}_{i-1} \setminus i$ if the updates were $X_i \leftarrow X_{i-1}$, and $Y_i \leftarrow Y_{i-1} \setminus i$ and $\mathrm{OPT}_i = \mathrm{OPT}_{i-1}$ if the updates were $X_i \leftarrow X_{i-1} \cup i$, and $Y_i \leftarrow Y_{i-1}$. So,

$$
\begin{aligned}
\mathbb{E}[\nu(\mathrm{OPT}_{i-1}) - \nu(\mathrm{OPT}_i)] &= \frac{b_i}{a_i + b_i}[\nu(\mathrm{OPT}_{i-1}) - \nu(\mathrm{OPT}_{i-1} \setminus i)] \\
&\leq \frac{b_i}{a_i + b_i}[\nu(X_{i-1} \cup i) - \nu(X_{i-1}) + (n-i)\gamma] \\
&\leq \frac{a_i b_i}{a_i + b_i} + (n-i)\gamma/2.
\end{aligned}
$$

The first inequality follows from weak submodularity since $(\text{OPT}_{i-1} \setminus i) \subseteq Y_{i-1} \setminus (X_{i-1} \cup i)$ which again has cardiality $n - i$. Since the same bound holds in either case we may now bound

$$
\begin{aligned}
\mathbb{E}[\nu(\text{OPT}_{i-1}) - \nu(\text{OPT}_i)] &\leq \frac{a_i b_i}{a_i + b_i} + (n - i)\gamma/2 \\
&\leq \frac{1}{2} \frac{a_i^2 + b_i^2}{a_i + b_i} + (n - i)\gamma/2 \\
&= \frac{1}{2}\left\{ \mathbb{E}[\nu(X_i) - \nu(X_{i-1}) + \nu(Y_i) - \nu(Y_{i-1}))] + (n-i)\gamma/2 \right\} + (n-i)\gamma/2 \\
&= \frac{1}{2}\mathbb{E}[\nu(X_i) - \nu(X_{i-1}) + \nu(Y_i) - \nu(Y_{i-1}))] + \frac{3}{4}(n-i)\gamma,
\end{aligned}
$$

where the second inequality uses the fact that $2ab \leq a^2 + b^2$ for all $a, b \in \mathbb{R}$.

$\square$

*Proof of Theorem 32.* Summing Lemma 33 over all $i$ yields,

$$
\sum_{i=1}^n \mathbb{E}[\nu(\text{OPT}_{i-1}) - \nu(\text{OPT}_i)] \leq \sum_{i=1}^n \left\{ \frac{1}{2}\mathbb{E}[\nu(X_i) - \nu(X_{i-1}) + \nu(Y_i) - \nu(Y_{i-1})] + \frac{3}{4}(n-i)\gamma \right\}.
$$

Both sides are telescoping sums. Recalling that $\text{OPT}_0 = \text{OPT}$ and $\text{OPT}_n = X_n = Y_n$ the sums collapse down to,

$$
\begin{aligned}
\mathbb{E}[\nu(\text{OPT}) - \nu(X_n)] &\leq \mathbb{E}[\nu(X_n) - \nu(X_0) + \nu(Y_n) - \nu(Y_0)] + \frac{3}{8}(n-1)n\gamma \\
&\leq \frac{1}{2}\mathbb{E}[\nu(X_n) + \nu(Y_n)] + \frac{3}{8}(n-1)n\gamma \\
&= \mathbb{E}[\nu(X_n)] + \frac{3}{8}(n-1)n\gamma,
\end{aligned}
$$

which rearranges to,

$$
\mathbb{E}[\nu(X_n)] \geq \frac{1}{2}\nu(\text{OPT}) - \frac{3}{16}(n-1)n\gamma.
$$

$\square$

## C.4 Greedy guarantees for increasing weak submodular functions

We call a set function $\nu$ *increasing* if $\nu(S) \leq \nu(T)$ whenever $S \subseteq T$. We can greedily obtain an estimate for $\max_{S:|S|\leq k} \nu(S)$ by setting $S_0 = \varnothing$ and recursively computing $S_\ell$ by adding $\arg\max_{i \in [n]} \nu(S_{\ell-1} \cup \{i\})$ to $S_{\ell-1}$ for $\ell = 1, \ldots, k$. For the remainder of this section we shall use OPT to denode an element of the optimal solution set $\arg\max_{S:|S|\leq k} \nu(S)$.

**Lemma 34.** *Let $\nu : 2^{[n]} \to \mathbb{R}_+$ be an increasing $\gamma$-weakly submodular function and $S_\ell \subseteq [n]$ be the set of size $\ell$ obtained by greedily optimizing $\nu$. Then for $k \leq n$ we have,*

$$
\nu(S_\ell) \geq (1 - e^{-\ell/k})\nu(OPT) - a
$$

*where $a = \frac{1}{2}k(k-1)\left\{1 - \left(1 - \frac{1}{k}\right)^\ell\right\}\gamma$. In particular, for $k = \ell$ we have,*

$$
\nu(S_k) \geq (1 - 1/e)\nu(OPT) - a.
$$

The bound is not vacuous so long as $\nu(\text{OPT}) = \max_{S:|S| \le k} \nu(S)$ grows as $O(k^2)$.

*Proof of Lemma 34.* Enumerate $\text{OPT} = \{v_1, \ldots, v_k\}$. Then for each $i < \ell$,

$$\nu(\text{OPT}) \le \nu(\text{OPT} \cup S_i) \tag{8}$$

$$= \nu(S_i) + \sum_{j=1}^{k} \left\{ \nu(S_i \cup \{v_1, \ldots, v_j\}) - \nu(S_i \cup \{v_1, \ldots, v_{j-1}\}) \right\} \tag{9}$$

$$= \nu(S_i) + \sum_{j=1}^{k} \nu(v_j \mid S_i \cup \{v_1, \ldots, v_{j-1}\}) \tag{10}$$

$$\le \nu(S_i) + \sum_{j=1}^{k} \left\{ \nu(v_j \mid S_i) + (j-1) \log \gamma \right\} \tag{11}$$

$$\le \nu(S_i) + \sum_{j=1}^{k} \left\{ \nu(S_{i+1}) - \nu(S_i) \right\} + \frac{1}{2} k(k-1) \log \gamma \tag{12}$$

$$\le \nu(S_i) + k\big(\nu(S_{i+1}) - \nu(S_i)\big) + \frac{1}{2} k(k-1) \log \gamma \tag{13}$$

where (8) follows from monotonoicity of $\nu$, (9) is a telescoping sum, (11) is obtained by bounding each term in the sum using Theorem 10, and (12) uses the fact that $S_{i+1}$ attains the maximal value of $\nu$ over all sets of size $i + 1$. Defining $\delta_i = \nu(\text{OPT}) - \nu(S_i)$ we have therefore obtained $\delta_i \le k(\delta_i - \delta_{i+1}) + \frac{1}{2} k(k-1) \log \gamma$. This rearranges to

$$\delta_{i+1} \le \left( 1 - \frac{1}{k} \right) \delta_i + \frac{1}{2} (k-1) \log \gamma.$$

Unrolling this recursive relation we find that,

$$\delta_\ell \le \left( 1 - \frac{1}{k} \right)^\ell \delta_0 + \frac{1}{2} (k-1) \sum_{i=0}^{\ell-1} \left( 1 - \frac{1}{k} \right)^i \log \gamma$$

$$\le e^{-\ell/k} \delta_0 + \frac{1}{2} k(k-1) \left\{ 1 - \left( 1 - \frac{1}{k} \right)^\ell \right\} \log \gamma.$$

Substituting back in the fact that $\delta_i = \nu(\text{OPT}) - \nu(S_i)$ and that $\nu(\varnothing) \ge 0$ and rearranging obtains the result.

$\square$

**Corollary 35.** *Let $\nu : 2^{[n]} \to \mathbb{R}$ be an increasing $\gamma$-weakly log-submodular function and $S_\ell \subseteq [n]$ be the set of size $\ell$ obtained by greedily optimizing $\nu$. Then for $k \le n$ we have,*

$$\nu(S_\ell) \ge e^{-a} \nu(OPT)^{(1 - e^{-\ell/k})}$$

*where $a = \frac{1}{2} k(k-1) \left\{ 1 - \left( 1 - \frac{1}{k} \right)^\ell \right\} \log \gamma$. In particular, for $k = \ell$ we have,*

$$\nu(S_k) \ge e^{-a} \nu(OPT)^{(1 - 1/e)}.$$

# D  Further experimental results

In this section we report the empirical mixing time results for different spectra on the positive semi-definite matrix $L$ as described in Section 6.

## (ii) $L$ has one big eigenvalue

(a)

(b)

Figure 3: Empirical mixing time analysis for sampling a ground set of size $n = 250$ and various cardinality constraints $d$, (a) the PSRF score for each set of chains, (b) the approximate mixing time obtained by thresholding at PSRF equal to $1.05$.

## (ii) $L$ has one big eigenvalue

(a)

(b)

Figure 4: Empirical mixing time analysis for sampling a set of size at most $d = 40$ for varying ground set sizes, (a) the PSRF score for each set of chains, (b) the approximate mixing time obtained by thresholding at PSRF equal to $1.05$.

**(iii) $L$ has a step in its spectrum**

(a)

(b)

Figure 5: Empirical mixing time analysis for sampling a ground set of size $n = 250$ and various cardinality constraints $d$, (a) the PSRF score for each set of chains, (b) the approximate mixing time obtained by thresholding at PSRF equal to $1.05$.

**(iii) $L$ has a step in its spectrum**

(a)

(b)

Figure 6: Empirical mixing time analysis for sampling a set of size at most $d = 40$ for varying ground set sizes, (a) the PSRF score for each set of chains, (b) the approximate mixing time obtained by thresholding at PSRF equal to $1.05$.

# E    Supplementary material on strongly log-concave polynomials and log-Sobolev inequalities

## E.1    Some theory for strongly log-concave polynomials

In the interests of completeness this section recalls some recent results that we rely on for our analysis. We shall call a polynomial $f$ *indecomposable* if it cannot be written as a sum $f = f_1 + f_2$ where $f_1, f_2$ are non-zero polynomials in disjoint sets of variables. In other words, the graph with nodes $\{i \mid \partial_i f \neq 0\}$ and edges $\{(i, j) \mid \partial_i \partial_j f \neq 0\}$ is connected. To state the next theorem we must introduce some more notation: for $\alpha \in \mathbb{N}^n$ define $|\alpha| = \sum_{i=1}^{n} \alpha_i$.

The following theorem provides checkable conditions for proving a polynomial is SLC.

**Theorem 36.** *[4] Let $f \in \mathbb{R}_+[z_1, \ldots, z_n]$ be homogeneous and of degree $d \geq 2$. If the following two conditions hold, then $f$ is SLC,*

1. *For all $\alpha \in \mathbb{N}^n$ with $|\alpha| \leq d - 2$ the polynomial $\partial^\alpha f$ is indecomposable.*

2. *For all $\alpha \in \mathbb{N}^n$ with $|\alpha| = d - 2$ the polynomial $\partial^\alpha f$ is log-concave on $\mathbb{R}^n_+$.*

**Theorem 37.** *[10] If $f$ homogeneous and SLC then,*

$$f(z)\partial_i\partial_j f(z) \le 2\left(1 - \frac{1}{d}\right)\partial_i f(z)\partial_j f(z)$$

*for all $z \in \mathbb{R}^n_+$.*

**Lemma 38.** *[4] Suppose $f \in \mathbb{R}[z_1, \ldots, z_n]$ is homogenous and SLC and let $T : \mathbb{R}^n \to \mathbb{R}^n$ be such that if $z \in \mathbb{R}^n_+$ then so is $Tz$. Then $f \circ T \in \mathbb{R}[z_1, \ldots, z_n]$ is also SLC.*

**Lemma 39.** *Suppose $f \in \mathbb{R}[z_1, \ldots, z_n]$ is homogenous and $a \in \mathbb{R}_+$ such that $f(a) \ne 0$, and set $Q = \nabla^2 f \mid_{z=a}$. Then the following are equivalent,*

1. *$f$ is log-concave at $z = a$,*

2. *$(a^\top Q a)Q - (Qa)(Qa)^\top$ is negative semidefinite,*

3. *$Q$ has at most one positive eigenvalue.*

The equivalence of the first two statements is proven in [4]. The third is not proven by Anari et al. but is a simple consequence of a result from [4]. Finally we note the following characterization of the support of a homogenous SLC distribution. It is a special case of a result due to Brändén and Huh ([10], Theorem 7.1). Let $e_i$ denote the $i$th standard basis vector. We shall call a set $J \subseteq \mathbb{N}^n$ *M-convex* if for any $\alpha, \beta \in J$ and any index $i$ such that $\alpha_i > \beta_i$, there exists an index $j$ such that $\alpha_j < \beta_j$ and $\alpha - e_i + e_j \in J$.

**Theorem 40.** *[10] Suppose $f = \sum_{|\alpha|=d} c_\alpha z^\alpha$ is SLC. Then the support $\{\alpha \in \mathbb{N}^d \mid c_\alpha \ne 0\}$ of $f$ is an M-convex set.*

## E.2 Log-Sobolev inequalities for bounding mixing times

Throughout this section we consider a finite. state space $\mathcal{X}$ and all distributions, kernels, and other functions mentioned will be defined on it. We define the Dirichlet form of a function $f$ with respect to a Markov chain to be

$$\mathcal{E}(f, f) = \frac{1}{2}\sum_{x,y \in \mathcal{X}}\left|f(x) - f(y)\right|^2 Q(x,y)\pi(x).$$

We also introduce the following entropy-like quantity

$$\mathcal{L}(f) = \sum_{x \in \mathcal{X}} f(x)^2 \log\left(\frac{f(x)^2}{\|f\|_2^2}\right)\pi(x).$$

**Definition 41.** The log-Sobolev constant of the Markov chain $(Q, \pi)$ is the largest $\alpha > 0$ such that

$$\alpha\mathcal{L}(f) \le \mathcal{E}(f, f)$$

for all $f$.

**Theorem 42.** *[19] Let $(Q, \pi)$ and $(Q', \pi')$ with Dirichlet forms and log-Sobolev constants $\mathcal{E}, \alpha$ and $\mathcal{E}', \alpha'$ respectively. Suppose there exists constants $a, A > 0$ such that*

$$\mathcal{E}' \le A\mathcal{E} \quad and \quad \pi \le a\pi'.$$

*Then*

$$\alpha' \le aA\alpha.$$

So, suppose you had a chain of interest $(Q, \pi)$ and happened to already know what $\alpha'$ was for some other chain $(Q', \pi')$, then by determining the constants $a, A > 0$ one obtains a lower bound on $\alpha$. This immediately yields an upper bound on the mixing time of $(Q, \pi)$.

## Footnotes

[2]This result was independently discovered by Brändén and Huh [10].