[Reviews · NeurIPS 2019]

Reviewer 1



The MCMC sampling is performed by homogenizing the distribution and using existing mixing results for the homogeneous case. This makes the argument simple and the bound is also weak, involving 2^d. This is the main weakness of the paper. Questions: * In which cases the assumptions of theorems 3,4 hold? In addition to SLC, they have some matroid related assumptions. Since these results intend to demonstrate the power of the SLC class, these should be discussed in more detail. * How the diversity related \alpha enters the mixing bounds? It seems that the bound depends very weakly on \alpha only through \nu(S_0). Edit following author's response: I'm inclined to keep my score of 6. This is due to following reasons: 1) I still find the theoretical contribution ok but not particularly strong, given existing results. As mentioned in the review, it is a weak, unpractical bound, and the proof, in itself, does not provide particular mathematical novelty. 2) The claims that "in practice the mixing time is even better" are not nearly sufficiently supported by the experiments, and therefore the evidence provided to practitioners is very limited. 3) My question regarding dependence on $\alpha$ was not answered in a satisfactory manner. I would expect a more explicit dependence on $\alpha$, since with higher diversity the problem should be more complicated. If this is not reflected in the bounds, it means the bounds are very loose.

Reviewer 2



Originality: The two algorithms presented in the paper are basically applications of the existing algorithms, namely, Metropolis Hastings sampler and distorted greedy algorithm. However, the paper presents nice new theoretical results which partly guarantee that the proposed algorithms work well. The paper clearly explains how this work is related to existing ones and provides an adequate number of citations of the related works. Quality: The quality of the paper is generally high. The authors have done careful work in terms of both theory and experiment. In particular, I think that the theoretical results of the paper are interesting and have a great value. Clarity: The paper is clearly written and well-organized. Although the paper includes some mathematical terminology which many participants of the conference might not be familiar with, I imagine that the nice presentation of the paper enables such participates to understand at least intuitively the contents of the paper. Regarding the presentation of the paper, I have only the following minor comments: (a) line 25: Is the word "conxevity" correct? (b) p.3, last line: The range of $y$ or the domain of $H_k f$ should be given. (c) line 175: ov ===> of (d) lines 222 and 223: It is not clear what 'all examples discussed above' refer to. (e) lines 250-252: The definition of OPT should be given. (f) line 255: appling ===> applying Significance: I think that the algorithms and theoretical results of the paper are important. The presented algorithms are based on the existing ones, but they are potentially useful for practitioners. The theoretical results related to the algorithms are very nice. The conclusions induced from the experimental results do not seem particularly significant, but it is helpful to find how one of the presented algorithms works under some settings of the size and parameters. EDIT: The authors have carefully responded to my comments, and I am satisfied with their responses. I think this is a nice paper and I will keep my good score.

Reviewer 3



• Originality: the authors are very clear on what is prior work and what their contributions are. Their algorithmic contributions are both (i) clearly novel and (ii) significant improvements over naïve baselines. • Quality: the theorems and proofs are written clearly and concisely. The authors cover all the natural sub-topics that ought to be covered when connecting two fields like this: both (i) on the algorithmic side, with sampling and mode-finding algorithms, and (ii) analyzing the theoretical properites thereof. • Clarity: the paper was very well-organized and clear. Significance: while strong empirical results are left to future work, the algorithmic and theoretical contributions are significant ant stand on their own.

[Author Response · NeurIPS 2019]

Sincere thanks to all the reviewers for the time and effort spent on reviewing our work.

**Response to reviewer 1:** Thank you very much for your thoughtful feedback.

**The role of diversity:** Submodularity is known as a diversity inducing property for probability distributions. Our
contribution in establishing a weak submodular property therefore sheds light on this. Furthermore, $\alpha$ is a tunable
parameter that allows the user to adjust the level of diversity desired: as $\alpha \to 0$ one obtains increasingly uniform
distributions (uniform being the least diversity inducing distribution). Whilst outside the scope of our results, making
$\alpha \gg 1$ results in an extreme preference for diversity: probability mass concentrates at the mode of of the original
distribution. That is, the "most diverse" set. We can make these connections more clear in the final copy of the paper.
More broadly SLC is an extension of SR which is well known to have some strong negative dependence properties.

**Mixing time bound $2^d$:** Although the mixing time bound is $2^d$, $d$ is a user-chosen parameter and may often be quite
small (much smaller than $n$). Furthermore it is a substantial improvement on $2^n$ that one obtains by a more naïve
analysis. Indeed the user would do well to consider mixing time as one of their desiderata when selecting $d$. The
experiments section provides evidence that the mixing time bound could in fact be tightened. The difficulty of proving
mixing time bounds for samplers is notorious and usually omitted all together, so we consider our contribution here to
be important nonetheless.

**Mixing time bound argument simplicity:** Whilst the idea of "lifting" to the homogenous case and using existing
results may be attractively simple, how to actually do this lifting in a way that yields an algorithm that empirically
works and has guarantees turned out to be a thorny question. In particular, a simpler proof based on closure under ho-
mogenization that applies for stricter classes of distributions (e.g., Strongly Rayleigh) does not apply here. Overcoming
this difficulty required much more theoretical work. We view it as a virtue that this work combines a simple high-level
argument with a careful and precise analysis to obtain theoretical guarantees.

**Matroid assumptions:** To the absolute best of our knowledge there are no known examples of non-homogenous SLC
distributions that have support not equal to the set of independent sets of a matroid. Indeed we consider it a theoretical
direction for future work to show that this is always true. In [10] it is shown that for the homogeneous case, the support is
the set of bases of a matroid. This, and other theory developed in that work suggest a fundamental connection between
SLC polynomials and matroids. We are very happy to include a discussion of this in the final copy.

**Diversity in mixing time bound:** Indeed you are correct, the $\alpha$ constant only enters the bound through $\nu(S_0)$. However,
bounds that depend on the distribution only through the starting point probability in this way are the norm in related
works [e.g. 2,5,40] and are a direct consequence of the deep theory of Markov chain mixing developed in [18,19].

**Response to reviewer 2:** Thank you very much for your kind words, useful suggestions, and thorough inspection.

**Metropolis-Hastings novelty:** Whilst the sampler is most certainly a MH sampler, the key question addressed
throughout the sampling section is: *what is the right proposal distribution?* Our contribution was the choosing of the
correct homogenized distribution to "lift" to, and to then demonstrate that with this choice we can prove mixing time
bounds on the algorithm. This was very much a non-trivial question that required some careful analysis to answer.

**Empirical contribution:** The experiments section provides empirical evidence that the mixing time is faster than our
bound. They are designed to point to an interesting and challenging direction for future work: refining the mixing time
bound. The difficulty of proving mixing time bounds for samplers is notorious and usually omitted all together, so we
consider our theoretical contributions here to be important nonetheless.

**Mathematical terminology:** We believe the terminology most likely to cause problems to be the references to matroid
theory. In response to Reviewer 1 we will include a discussion of these assumptions in the final copy. We can also take
that opportunity to reassure readers unfamiliar with matroids that they play only a minimal technical role and can be
largely disregarded by all except those who want to delve into the theory of SLC distributions.

**Minor comments:** Thank you for pointing out typos. The definition of OPT was given in the first paragraph of Section
5. For points (b),(d) we will make these points clear in the final copy.

**Response to reviewer 3:** Thank you very much for your supportive feedback and thoughts, we greatly appreciate it.

**Empirical contributions:** Whilst not being large scale tests by any means, they serve a particular function in this paper
to point to an interesting and challenging direction for future work: refining the mixing time bound.

**Bridging fields:** We are pleased that you recognize this contribution. Indeed, we view this as a main contribution of the
paper; it opens up SLC modelling as a new line of work to the ML community.

**Further empirical work:** We decided to leave the careful and thorough application of the new methods introduced to
future work. We see this as an important topic in its own right and worthy of a separate piece of work.

[Meta-Review · NeurIPS 2019]

This paper provides sampling and mode-finding algorithms for strongly log-concave distributions, along with some theoretical results, e.g., mixing-time analysis. The reviewers were unanimous in their vote to accept. Authors are encouraged to revise with respect to reviewer comments.